# Weighted Risk Invariance: Domain Generalization under Invariant Feature Shift

**Gina Wong**                                                    *gwong15@jhu.edu*
*Department of Computer Science*
*Johns Hopkins University*

**Joshua Gleason**                                              *gleason@umd.edu*
*Department of Electrical and Computer Engineering*
*University of Maryland, College Park*

**Rama Chellappa**                                              *rchella4@jhu.edu*
*Department of Electrical and Computer Engineering*
*Johns Hopkins University*

**Yoav Wald**                                                   *yoav.wald@nyu.edu*
*Center for Data Science*
*New York University*

**Anqi Liu**                                                    *aliu.cs@jhu.edu*
*Department of Computer Science*
*Johns Hopkins University*

**Reviewed on OpenReview:** *https://openreview.net/forum?id=WyPKLWPYsr*

## Abstract

Learning models whose predictions are invariant under multiple environments is a promising approach for out-of-distribution generalization. Such models are trained to extract features $X_{\text{inv}}$ where the conditional distribution $Y \mid X_{\text{inv}}$ of the label given the extracted features does not change across environments. Invariant models are also supposed to generalize to shifts in the marginal distribution $p(X_{\text{inv}})$ of the extracted features $X_{\text{inv}}$, a type of shift we call an *invariant covariate shift*. However, we show that proposed methods for learning invariant models underperform under invariant covariate shift, either failing to learn invariant models—even for data generated from simple and well-studied linear-Gaussian models—or having poor finite-sample performance. To alleviate these problems, we propose *weighted risk invariance* (WRI). Our framework is based on imposing invariance of the loss across environments subject to appropriate reweightings of the training examples. We show that WRI provably learns invariant models, i.e. discards spurious correlations, in linear-Gaussian settings. We propose a practical algorithm to implement WRI by learning the density $p(X_{\text{inv}})$ and the model parameters simultaneously, and we demonstrate empirically that WRI outperforms previous invariant learning methods under invariant covariate shift.

## 1 Introduction

Although traditional machine learning methods can be incredibly effective, many are based on the *i.i.d. assumption* that the training and test data are independent and identically distributed. As a result, these methods are often brittle to distribution shift, failing to generalize training performance to *out-of-distribution* (OOD) data (Torralba & Efros, 2011; Beery et al., 2018; Hendrycks & Dietterich, 2019; Geirhos et al., 2020). Distribution shifts abound in the real world: as general examples, we encounter them when we collect test data under different conditions than we collect the training data (Adini et al., 1997; Huang et al., 2006), or

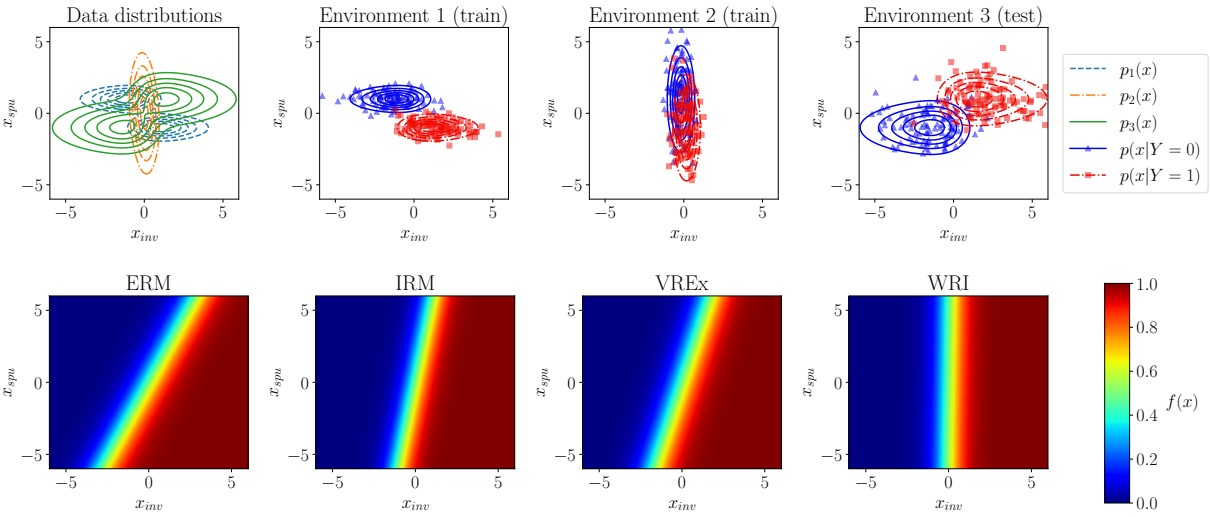

**Figure 1:** Top row: a Gaussian setup exhibiting both heteroskedasticity and invariant covariate shift. Note that for invariant features $x_{inv}$, $p(y|x_{inv})$ is the same across environments, while $p(y|x_{spu})$ changes across environments, following our causal model. Bottom row: we visualize the performance of different algorithms under this setup, and find the WRI objective recovers a more invariant predictor than the other algorithms. Specifically, the WRI predictor bases its predictions more on the invariant features $x_{inv}$ and less on the spurious features $x_{spu}$, so it learns a decision boundary that is vertical in this case. The full parameters for this simulation can be found in Appendix E.

when we train on synthetic data and evaluate on real data (Chen et al., 2020; Beery et al., 2020). Learning to generalize under distribution shift, then, is a critical step toward practical deployment.

To achieve generalization, a typical setup is to use multiple training environments with the hope that, if the model learns what is invariant across the training environments, it can leverage that invariance on an unseen test environment as well. An extensive line of research therefore focuses on learning statistical relationships that are invariant across training environments (Ganin et al., 2016; Tzeng et al., 2017; Li et al., 2021). Of these, many works focus on learning an invariant relationship between the data $X$ and labels $Y$, as such relationships are posited to result from invariant causal mechanisms (Pearl, 1995; Schölkopf et al., 2012; Peters et al., 2016; Rojas-Carulla et al., 2018). Applying this idea to representation learning, several works have found that learning features $\Phi(X)$ such that the label conditional distribution $p(Y|\Phi(X))$ is invariant across training environments is an effective method for improving generalization to unobserved test environments (Arjovsky et al., 2019; Krueger et al., 2021; Wald et al., 2021; Eastwood et al., 2022).

Predictors that rely on conditionally invariant features can be seen as an optimal solution to the problem of OOD generalization (Koyama & Yamaguchi, 2020). However, we demonstrate that existing methods for finding these predictors can struggle or even fail when the conditionally invariant features undergo covariate shift, a type of shift we call *invariant covariate shift*. We show that invariant covariate shift introduces two types of challenges: first, it leads risk invariance objectives (such as Krueger et al. (2021)) to incorrectly identify the conditionally invariant features in a heteroskedastic setting,[1] and second, it adversely impacts the sample complexity (Shimodaira, 2000) of recovering an optimal invariant predictor (in the style of Arjovsky et al. (2019)). For an illustration of these failure cases, see the learned predictors for VREx and IRM respectively in Figure 1.

In this work, we demonstrate how to mitigate the effects of invariant covariate shift by accounting for the underlying structure of the data in our learning process. We propose Weighted Risk Invariance (WRI), where we enforce invariance of a model's reweighted losses across training environments. We show that under appropriate choices of weights, models satisfying WRI provably achieve OOD generalization by learning to reject spurious correlations, under a common model for learning under spurious correlations (Arjovsky

---

[1]That is, where classification is more difficult in some instances than others.

et al., 2019; Rosenfeld et al., 2020; Wald et al., 2021). To learn models that satisfy WRI in practice, we introduce a method that alternates between estimation of reweightings (corresponding to densities in our implementation) and optimization of a prediction loss penalized by an empirical estimate of WRI. We show that solving for WRI recovers invariant predictors, even in the difficult setting of heteroskedasticity and invariant covariate shift, where popular previously proposed invariant learning principles fail (Figure 1). Finally, we show that under invariant covariate shift, the WRI solution outperforms widely used baselines.

In summary, our main contributions are:

- We introduce an invariant learning method that uses weighted risk invariance to tackle invariant covariate shift. We formally show that enforcing weighted risk invariance recovers an invariant predictor under a linear causal setting (Prop. 1, Thm. 1).

- We propose WRI, an optimization problem for enforcing weighted risk invariance, and provide an algorithm to solve WRI in practice via alternating minimization (§3.4).

- We verify the efficacy of our approach with experiments on simulated and real-world data. We find that, under invariant covariate shift, our method outperforms widely used baselines on benchmarks like ColoredMNIST and the DomainBed datasets. Moreover, our learned densities report when conditionally invariant features are rare in training, making them useful for downstream tasks like OOD detection (§4).

## 2 Preliminaries

### 2.1 Domain generalization

Domain generalization was first posed (Blanchard et al., 2011; Muandet et al., 2013) as the problem of learning some invariant relationship from multiple training domains/environments $\mathcal{E}_{tr} = \{e_1, \ldots, e_k\}$ that we assume to hold across the set of all possible environments we may encounter $\mathcal{E}$. Each training environment $E = e$ consists of a dataset $D^e = \{(\mathbf{x}_i^e, y_i^e)\}_{i=1}^{n_e}$, and we assume that data pairs $(\mathbf{x}_i^e, y_i^e)$ are sampled i.i.d from distributions $p_e(X^e, Y^e)$ with $\mathbf{x} \in \mathcal{X}$ and $y \in \mathcal{Y}$.[2]

For loss function $\ell : \hat{\mathcal{Y}} \times \mathcal{Y} \to \mathbb{R}$, we define the statistical *risk* of a predictor $f : \mathcal{X} \to \hat{\mathcal{Y}}$ over an environment $e$ as

$$\mathcal{R}^e(f) = \mathbb{E}_{p_e(X^e, Y^e)} \left[ \ell(f(\mathbf{x}^e), y^e) \right], \tag{1}$$

and its empirical realization as

$$\frac{1}{|D^e|} \sum_{i=1}^{|D^e|} \ell(f(\mathbf{x}_i^e), y_i^e). \tag{2}$$

Empirical Risk Minimization (ERM) (Vapnik, 1991) aims to minimize the average loss over all of the training samples

$$\mathcal{R}_{ERM}(f) = \frac{1}{|\mathcal{E}_{tr}|} \sum_{e \in \mathcal{E}_{tr}} \mathcal{R}^e(f). \tag{3}$$

Unfortunately, ERM fails to capture distribution shifts across training environments (Arjovsky et al., 2019; Krueger et al., 2021), and so can fail catastrophically depending on how those distribution shifts extend to the test data. (Previous works point to cases where ERM successfully generalizes, but the reason is not well understood (Vedantam et al., 2021; Gulrajani & Lopez-Paz, 2020), and its success does not hold across all types of distribution shifts (Ye et al., 2022; Wiles et al., 2021).) In our work, we therefore aim to learn a predictor that captures the underlying causal mechanisms instead, under the assumption that such mechanisms are invariant across the training and test environments.

### 2.2 Our causal model

We assume that input $X$ contains both causal/invariant and spurious components such that there exists a deterministic mechanism to predict the label $Y$ from invariant components $X_{\text{inv}}$, while the relationship

---

[2]When the context is clear, we write the data-generating random vector $X$, output random variable $Y$, and their realizations without the environment superscript.

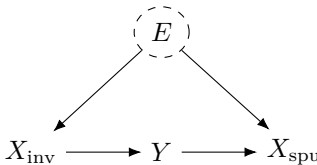

**Figure 2:** Causal graph depicting our data-generating process. The environment $E$ is dashed to emphasize it takes on unobserved values at test time.

between spurious components $X_{\text{spu}}$ and label $Y$ depends on the environment. More explicitly, we say that $Y \not\perp\!\!\!\perp E \mid X_{\text{spu}}$. The observed features $X$ are a function of the invariant and spurious components, or $X = g(X_{\text{inv}}, X_{\text{spu}})$, which we assume is injective so that there exists $g^\dagger(\cdot)$ that recovers $X_{\text{spu}}, X_{\text{inv}}$ almost surely. We represent this data-generating process with the causal graph in Figure 2. From the graph, we observe that $p_e(Y \mid X_{\text{inv}}) = p_{\tilde{e}}(Y \mid X_{\text{inv}})$ for any $e, \tilde{e} \in \mathcal{E}$, i.e. it is fixed across environments. On the other hand, the distribution $p_e(X_{\text{spu}} \mid Y)$ is not fixed across various $e \in \mathcal{E}$ and we assume it may change arbitrarily. We summarize these assumptions below.

**Assumption 1** (Causal Prediction Coupling). A causal prediction coupling is a set of distributions $\mathcal{P} = \{p_e\}_e$, where each $p_e \in \mathcal{P}$ is a distribution over real vectors $X = (X_{\text{inv}}, X_{\text{spu}})$ and target $Y$ that decomposes according to

$$p_e(X, Y) = p_e(X_{\text{inv}})p(Y \mid X_{\text{inv}})p_e(X_{\text{spu}} \mid Y).$$

We call $X_{\text{inv}}$ and $X_{\text{spu}}$ the invariant and spurious components accordingly. The distribution $p(Y \mid X_{\text{inv}})$ is fixed across all members of $\mathcal{P}$.

Intuitively, we seek a robust predictor $f$ where $\mathcal{R}^e(f)$ does not grow excessively high for any $p_e \in \mathcal{P}$. Let us focus on the square and logistic losses, so an optimal model with respect to $p_e$ predicts $Y$ according to its conditional expectation $\mathbb{E}_{p_e}[Y \mid X]$. Focusing on couplings where $p_e(X_{\text{spu}} \mid Y)$ can undergo large changes, we observe that a guarantee on the worst case loss can only be obtained when $f(\mathbf{x})$ does not depend on $X_{\text{spu}}$. This can be shown formally for the case of linear regression, by showing that any model $f(\mathbf{x}) = \mathbf{w}^\top \mathbf{x}$ where $\mathbf{w}_{\text{spu}} \neq 0$ can incur an arbitrarily high loss under some $p_e \in \mathcal{P}$ (Arjovsky et al., 2019; Rosenfeld et al., 2020). Thus, we wish to learn a predictor that satisfies $f(X) = f(X_{\text{inv}})$ almost everywhere.

Restricted to such predictors, we observe two things. First, there is a single model that is simultaneously optimal for all $\mathcal{P}$, and this model is the predictor we would like to learn. Second, the worst-case loss of this predictor is controlled by the loss of the most difficult instances in terms of $X_{\text{inv}}$, since $p_e(X_{\text{inv}})$ can shift mass to these instances where the predictor incurs the highest loss. Following these arguments, we arrive at definitions of a spurious-free predictor and an invariant predictor.

**Definition 1.** A measurable function $f : \mathcal{X} \to \hat{\mathcal{Y}}$ is a *spurious-free predictor* with respect to a causal prediction coupling $\mathcal{P}$ if $f(X) = f(X_{\text{inv}})$ almost everywhere. Considering the square or logisitic losses, we call the optimal spurious-free predictor $f^*(\mathbf{x}) = \mathbb{E}[Y \mid X_{\text{inv}} = \mathbf{x}_{\text{inv}}]$ the *invariant predictor* for short.

Invariant predictors, or representations, are usually defined as models that satisfy $\mathbb{E}_{p_e}[Y \mid f(X)] = \mathbb{E}_{p_{e'}}[Y \mid f(X)]$ for any $e, e' \in \mathcal{E}'$ (Arjovsky et al., 2019).[3] For the settings we study in this paper, the invariant predictor as defined in Definition 1 satisfies this property.

## 2.3 Invariant learning under covariate shift

Causal predictors assume there exist features $X_{\text{inv}} = \Phi(X)$ such that the mechanism that generates the target variables $Y$ from $\Phi(X)$ is invariant under distribution shift; in other words, the conditional distribution $p(Y|\Phi(X))$ is invariant across environments (Schölkopf et al., 2012; Peters et al., 2016). This is similar to covariate shift, where the conditional distribution $p(Y|X)$ is assumed to be the same between training and test while the training and test distributions $p_{tr}(X)$ and $p_{te}(X)$ differ. However, while the covariate

---

[3]Demanding equality in distribution and not just expectation is also common.

shift assumption refers to distribution shift in the covariates $X$, we study the case of covariate shift in the conditionally invariant features $X_{\text{inv}}$, which we call *invariant covariate shift*. For simplicity, we will also call conditionally invariant features simply *invariant* when the context is clear, using terminology consistent with previous works (Liu et al., 2021a; Ahuja et al., 2021).

As a motivating example, consider the case where we have patient demographics $X$ and diagnostic information $Y$ from multiple hospitals. Some demographic features, like whether a patient smokes or not, may have a direct influence on a diagnosis like lung cancer. We can model these features as invariant features $X_{inv}$. Under invariant covariate shift, although the distribution of smokers $p(X_{inv})$ can differ from hospital to hospital, the relationship $p(Y|X_{inv})$ remains the same. Unlike regular covariate shift, which applies to the entire covariate, the assumption here is more targeted: under regular covariate shift, we assume that the conditional distribution given all the patient demographics $p(Y|X)$ is the same across hospitals. In contrast, the assumption that only $p(Y|X_{inv})$ is the same across hospitals is more realistic.

Several popular works search for proxy forms of invariance in an attempt to recover the invariant features from complex, high-dimensional data. Methods along this line include Risk Extrapolation (REx) (Krueger et al., 2021) and Invariant Risk Minimization (IRM) (Arjovsky et al., 2019). Unfortunately, these methods are adversely impacted by invariant covariate shift, as we demonstrate in §3.3. We will see there are two types of challenges, namely: (1) the failure to learn an invariant predictor, even given infinite samples—this is where methods like REx fail, even though they have favorable performance when there is no invariant covariate shift; and (2) finite sample performance, where methods like IRM underperform when there is invariant covariate shift, even though they are guaranteed to learn an invariant predictor as the amount of data grows to infinity.

A classical approach to counteract the effects of covariate shift is importance weighting (Shimodaira, 2000), where the training loss is reweighted with a ratio of test to training densities so that training better reflects the test distribution. Generalizing this concept, Martin et al. (2023) finds there are many choices of weights that can help align the training and test distributions; that is, there exist many choices of weights $\alpha(\mathbf{x})$ and $\beta(\mathbf{x})$ such that

$$\mathbb{E}_{p_{te}(X,Y)}\left[\alpha(\mathbf{x})\ell(f(\mathbf{x}), y)\right] = \mathbb{E}_{p_{tr}(X,Y)}\left[\beta(\mathbf{x})\ell(f(\mathbf{x}), y)\right]. \tag{4}$$

Note that this equation reduces to importance weighting under *density ratio weighting*, or when $\beta(\mathbf{x}) = p_{te}(\mathbf{x})/p_{tr}(\mathbf{x})$ and $\alpha(\mathbf{x}) = 1$. Under this more general framework, *density weighting*, where $\beta(\mathbf{x}) = p_{te}(\mathbf{x})$ and $\alpha(\mathbf{x}) = p_{tr}(\mathbf{x})$, achieves a similar effect.

Our work studies the question of how to adapt reweighting to the case of invariant learning. Unfortunately, a naive choice of weights, where the covariates of one environment are reweighted to match the distribution of the other (i.e. importance weighting), will not provide the desired result in the scenario of invariant learning.[4] In what follows, we show that for reweighting to be effective in this setting, it needs to explicitly account for the density of $X_{\text{inv}}$.

## 3 Density-aware generalization

### 3.1 Invariant feature density weighting

Consider choosing some set of weights $\alpha(\mathbf{x}_{\text{inv}}), \beta(\mathbf{x}_{\text{inv}})$ such that $\alpha(\mathbf{x}_{\text{inv}})p_{e_i}(\mathbf{x}_{\text{inv}}) = \beta(\mathbf{x}_{\text{inv}})p_{e_j}(\mathbf{x}_{\text{inv}})$ (Martin et al., 2023). This defines a pseudo-distribution $q(\mathbf{x}_{\text{inv}}) \propto \alpha(\mathbf{x}_{\text{inv}})p_{e_i}(\mathbf{x}_{\text{inv}})$ over the invariant features. It turns out that for spurious-free models $f(\mathbf{x})$, even when we reweigh the entire joint distribution $p_{e_i}(X, Y)$ instead of the marginal distribution over $X_{\text{inv}}$ alone, the expected loss will still be equal to that over $q(\mathbf{x}_{\text{inv}})$ (up to a normalizing constant). This also holds when we do the reweighting on another environment, which leads us to the following result proved in the appendix.

**Proposition 1.** *Let Assumption 1 hold over a set of environments $\mathcal{E} \subseteq \mathcal{P}$. If a predictor $f$ is spurious-free over $\mathcal{E}$, then for every pair of environments $e_i, e_j \in \mathcal{E}$, their weighted risks are equal if their respective weighting functions $\alpha$ and $\beta$ obey $\alpha(\mathbf{x}_{inv})p_{e_i}(\mathbf{x}_{inv}) = \beta(\mathbf{x}_{inv})p_{e_j}(\mathbf{x}_{inv})$.*

---

[4]This occurs because an invariant model on the original distribution may become not-invariant under the weighted data, where distributions of covariates are matched across environments. It is easy to see this with simple examples.

This result shows that this form of weighted risk invariance is a signature of spurious-free prediction, and it is easy to verify that the optimal invariant predictor also satisfies this invariance. We formalize this notion of invariance with the following definition.

**Definition 2.** We define weighted risk invariance between two environments $e_i$ and $e_j$ to mean that

$$\mathbb{E}_{p_{e_i}(X^{e_i}, Y^{e_i})} [\alpha(\mathbf{x}_{\text{inv}})\ell(f(\mathbf{x}), y)] = \mathbb{E}_{p_{e_j}(X^{e_j}, Y^{e_j})} [\beta(\mathbf{x}_{\text{inv}})\ell(f(\mathbf{x}), y)],$$

where $\alpha(\mathbf{x}_{\text{inv}})p_{e_i}(\mathbf{x}_{\text{inv}}) = \beta(\mathbf{x}_{\text{inv}})p_{e_j}(\mathbf{x}_{\text{inv}})$. Weighted risk invariance over a set of more than two environments means that weighted risk invariance holds for all pairwise combinations of environments in that set. When one of the weighting functions is a density ratio of the invariant features, i.e. when either $\alpha(\mathbf{x}_{\text{inv}}) = p_{e_j}(\mathbf{x}_{\text{inv}})/p_{e_i}(\mathbf{x}_{\text{inv}})$ and $\beta(\mathbf{x}_{\text{inv}}) = 1$, or $\alpha(\mathbf{x}_{\text{inv}}) = 1$ and $\beta(\mathbf{x}_{\text{inv}}) = p_{e_i}(\mathbf{x}_{\text{inv}})/p_{e_j}(\mathbf{x}_{\text{inv}})$, we say this is *density ratio weighting*. When the weighting functions are invariant feature densities, i.e. when $\alpha(\mathbf{x}_{\text{inv}}) = p_{e_j}(\mathbf{x}_{\text{inv}})$ and $\beta(\mathbf{x}_{\text{inv}}) = p_{e_i}(\mathbf{x}_{\text{inv}})$, we say this is *density weighting*.

However, weighted risk invariance alone is not enough to guarantee that a predictor is spurious-free. As an example, the case where $\alpha(\mathbf{x}_{\text{inv}}) = \beta(\mathbf{x}_{\text{inv}}) = 0$ allows any predictor to satisfy weighted risk invariance in a trivial manner, without excluding spurious features. To avoid these degenerate cases, we assume that the reweighted environments are in general position.[5] Then, as in previous work on principled methods for learning invariant models (Arjovsky et al., 2019; Krueger et al., 2021; Wald et al., 2021), we prove that predictors with weighted risk invariance discard spurious features $X_{\text{spu}}$ for OOD generalization under general position conditions. We show this for the case of linear regression, where we have the following data-generating process parameterized by $\mathbf{w}_{inv}^*$, $\sigma_y$, and $\mu_i$, $\Sigma_i$ for training environment $i \in [k]$:

$$
\begin{aligned}
Y &= \mathbf{w}_{\text{inv}}^{*\top} X_{\text{inv}} + \varepsilon, \ \varepsilon \sim \mathcal{N}(0, \sigma_y^2) \\
X_{\text{spu}} &= \mu_i Y + \eta, \ \eta \sim \mathcal{N}(0, \Sigma_i).
\end{aligned}
\tag{5}
$$

**Theorem 1.** *Consider a regression problem following the data generating process of equation 5. Let $\mathcal{E}$ be a set of environments with $|\mathcal{E}| > d_{spu}$ that satisfies general position. A linear regression model $f(\mathbf{x}) = \mathbf{w}^\top \mathbf{x}$ with $\mathbf{w} = [\mathbf{w}_{inv}, \mathbf{w}_{spu}]$ that satisfies weighted risk invariance w.r.t the squared loss must also satisfy $\mathbf{w}_{spu} = 0$. For density ratio weighting, general position holds with probability 1.*

This result shows that in a linear regression setting, solving for weighted risk invariance allows us to learn a predictor that discards spurious correlations to generalize across environments. The general position condition excludes cases such as $\alpha(\mathbf{x}_{\text{inv}}) = \beta(\mathbf{x}_{\text{inv}}) = 0$ that reduce to a degenerate set of equations. We further show that density ratio weighting achieves this non-degeneracy with probability 1, and we conjecture that this holds for other choices of weighting functions.

### 3.2 Finding an effective weighting function

Each weighting function introduces different challenges to the learning process. Density ratio weighting has the typical importance weighting limitation, where the support of the distribution in the numerator needs to be contained in the support of the distribution in the denominator. In addition, density ratio weights tend to have a high maximum and high variance, and as a result, lead to a high sample complexity—another known issue of importance weighting (Cortes et al., 2010).[6] Conversely, while the number of weights/constraints for the general form of weighted risk invariance is combinatorial in the number of environments, density ratio weighting only requires weights and constraints that are linear in the number of environments. (Intuitively, this is because we set a reference environment, weighted by $\alpha = 1$, and weight all other environments by a density ratio that aligns them with the reference environment; we discuss this in more detail in our proof of Theorem 1).

In practice, we find this benefit is not computationally significant when the number of environments is sufficiently small. Instead, we propose to use the invariant feature densities as the weighting functions. That is, for two environments $e_i$ and $e_j$, we use a weighted risk of

$$\mathcal{R}^{e_i, e_j}(f) = \mathbb{E}_{p_{e_i}(X^{e_i}, Y^{e_i})} \left[ p_{e_j}(\mathbf{x}_{\text{inv}})\ell(f(\mathbf{x}), y) \right]$$

---

[5]See Appendix A for the full definition of general position.

[6]See Appendix B for more discussion.

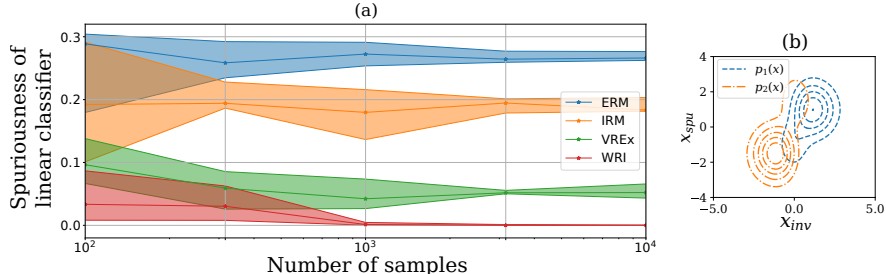

**Figure 3:** (a) We demonstrate the finite sample behavior of ERM, IRM, VREx, and WRI methods in the case where the distribution shift is large. We sample data from the distributions shown in (b). Even as the number of samples increases, ERM, IRM and VREx methods continue to select spurious classifiers while the WRI method quickly converges to an invariant classifier.

so that weighted risk invariance means $\mathcal{R}^{e_i, e_j}(f) = \mathcal{R}^{e_j, e_i}(f)$. There are several practical benefits to this approach. For one, density weighting does not impose requirements on the support of the environments, and allows for invariant learning over all regions of invariant feature support overlap.[7] And although density weighting uses a combinatorial number of constraints, the number of weighting functions we need to learn is still equal to the number of environments. As another benefit, learning the invariant feature densities gives us a signal indicating the distribution shift in invariant features across domains that is useful for downstream tasks. We therefore choose the weighting functions to be the invariant feature densities, and more precisely define weighted risk invariance as risk invariance with density weighting.

To enforce this weighted risk invariance, we propose to minimize the objective

$$\mathcal{R}_{WRI}(f) = \sum_{e \in \mathcal{E}_{tr}} \mathcal{R}^e(f) + \lambda \sum_{\substack{e_i \neq e_j, \\ e_i, e_j \in \mathcal{E}_{tr}}} (\mathcal{R}^{e_i, e_j}(f) - \mathcal{R}^{e_j, e_i}(f))^2. \tag{6}$$

The first term enforces ERM to ensure good average performance across the training environments. The second term enforces weighted risk invariance, so we call it the WRI penalty term. Hyperparameter $\lambda$ controls the weight of the WRI penalty.

### 3.3 Comparison to other invariant learning methods

**Comparison to REx**   Risk invariance across environments is a popular and practically effective method for achieving invariant prediction (Xie et al., 2020; Krueger et al., 2021; Liu et al., 2021b; Eastwood et al., 2022) that was formalized by REx (Krueger et al., 2021). Specifically, the VREx objective is a widely used approach of achieving risk invariance, where the variance between risks is penalized in order to enforce equality of risks:

$$\mathcal{R}_{VREx}(f) = \mathcal{R}_{ERM}(f) + \lambda_{VREx} \text{Var}(\{\mathcal{R}^{e_1}(f), \ldots, \mathcal{R}^{e_k}(f)\}). \tag{7}$$

The variance between risks is equal to the average mean squared error between risks, up to a constant. Note that the WRI penalty in equation 6 reduces to the VREx variance penalty under uniform weighting.

Under invariant covariate shift, risk invariance objectives can be misaligned with the goal of learning a predictor that only depends on $X_{\text{inv}}$. To illustrate this, consider a problem with heteroskedastic label noise (where some instances are more difficult to classify than others (Krueger et al., 2021)). Under this setting, invariant covariate shift results in a case where one training environment has more difficult examples than the other. Then an invariant predictor would obtain different losses on both environments, meaning it will not satisfy risk invariance. Figure 1 demonstrates this case, and shows that a risk invariance penalty (VREx) does not learn the invariant classifier. In contrast, weighted risk invariance holds when there is both heteroskedasticity and invariant covariate shift.

---

[7]Support overlap is necessary for invariant learning to meaningfully occur; this is a fundamental limitation of invariant learning in general (Ahuja et al., 2021), and violating this does not misalign the weighted risk objective. In non-overlapping regions, the weighted risk would simply be zero, causing the ERM objective to take over.

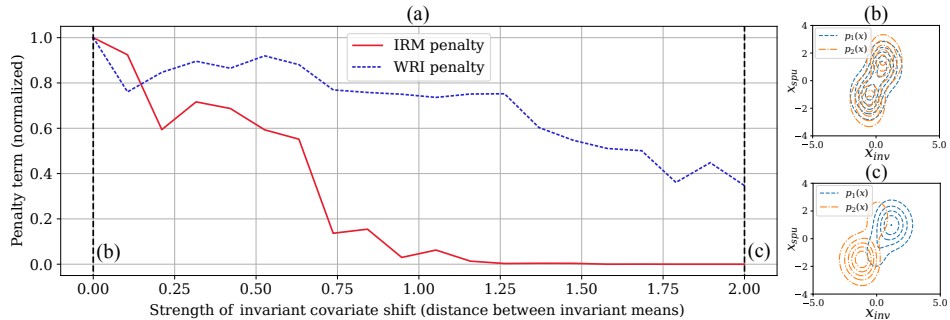

**Figure 4:** The WRI penalty is more sensitive to spurious classifiers than the IRM penalty under invariant covariate shift. (a) We start with two data distributions ($p_1$ and $p_2$) and increase the distance between their means from 0 to 2 in the invariant direction, essentially creating invariant covariate shift ranging in degree from the diagram in (b) to the diagram in (c). We then search through the set of challenging, slightly spurious classifiers with near-optimal ERM loss, so that we must rely on the invariant penalty to reject these classifiers. We plot how the minimum IRM and WRI penalties in these regions change as covariate shift increases.

**Comparison to IRM** IRM searches for the optimal predictor on latent space that is invariant across environments. This work generated many follow-up extensions (Ahuja et al., 2021; Lu et al., 2021), as well as some critical exploration of its failure cases (Rosenfeld et al., 2020; Ahuja et al., 2020; Kamath et al., 2021; Guo et al., 2021). The original objective is a bi-level optimization problem that is non-convex and challenging to solve, so its relaxation IRMv1 is more often used in practice:

$$\mathcal{R}_{IRM}(f) = \mathcal{R}_{ERM}(f) + \lambda_{IRM} \sum_{e \in \mathcal{E}_{tr}} \|\nabla_{w|w=1.0} \mathcal{R}^e(w \cdot f)\|_2^2. \tag{8}$$

Note that IRM uses the average gradient across the dataset to enforce invariance. This approach inherently introduces a sample complexity issue: when invariance violations are sparse or localized within a large dataset, the average gradient only indirectly accounts for these anomalies, requiring more samples to recognize the violations. Empirically, we observe that even for a toy example, the sample complexity for IRM is high under invariant covariate shift. We demonstrate this in Figure 3, where WRI converges to an invariant predictor while IRM does not. (Note that REx does not converge to an invariant predictor either—but we do not expect it to converge to an invariant predictor in this case, regardless of the amount of training data.)

In Figure 4, we demonstrate how the average loss-based WRI penalty is more sensitive to invariance violations than the average gradient-based IRM penalty. To create this figure, we start with a model that uses spurious features and observe how the IRM and WRI penalties for the model change as we introduce covariate shift to the invariant features. Ideally, the penalties should be consistently nonzero (regardless of the degree of shift) as the model is not invariant. We see that the IRM penalty changes quite significantly as the invariant covariate shift increases, while the change for WRI is less pronounced.

### 3.4 Implementation

Evaluating the WRI penalty requires weighting the loss by functions of the invariant features. However, we do not have access to the invariant features—otherwise the invariant learning problem would already be solved. In Appendix C, we empirically show that for a simple case of our regression setting where we can compute exact densities, optimizing for WRI on the joint features (rather than the invariant features) still leads to learning invariant feature weighting functions. Unfortunately, density estimation on high-dimensional data is notoriously difficult, and we often cannot compute exact density values in practice.

To tackle more realistic settings where we do not have exact density values, our approach must optimize for weighted risk invariance while also learning the feature densities. We therefore propose to use alternating minimization, an effective strategy for nonconvex optimization problems involving multiple interdependent terms. We propose to alternate between a WRI-regularized prediction model and environment-specific

density models. In one step, we keep the prediction model fixed and update a density model for each environment; in the alternating step, we keep all density models fixed and train the prediction model.

We do not offer any convergence guarantees for this approach. Even so, our experiments support that alternating minimization can recover an invariant predictor and density estimates effectively. Most of the experiments in our main body focus on using WRI to learn an invariant predictor. However, in Appendix D.3, we show that running alternating minimization on our regression setting recovers a close approximation of the ground truth densities (up to a normalizing constant). We also find that the learned densities are useful for predicting whether an input is part of the training invariant feature distributions or OOD in §4.1.

**Practical objective**    We want to avoid learning a representation where there is no overlap between environments (corresponding to zero weights for all data points), so we introduce a negative log penalty term to encourage high entropy and a large support for each environment. We observe empirically that this term encourages overlap between environments. When our density model matches the true distribution family of the features, the negative log term encourages a maximum likelihood estimate of the feature distribution parameters.

To make the dependence of the weighted risk on the invariant feature densities explicit, we write the associated densities $d^{e_j}$ as a parameter of the weighted risk $R^{e_i,e_j}(f; d^{e_j})$. As in §2.3, we denote the learned representation as $\Phi(\mathbf{x})$ so that $f = g \circ \Phi$ for some predictor $g$. The full loss function is

$$\mathcal{L}(f, \{d^{e_i}\}_{i=1}^k) = \sum_{e \in \mathcal{E}_{tr}} \mathcal{R}^e(f) + \lambda \sum_{\substack{e_i \neq e_j, \\ e_i, e_j \in \mathcal{E}_{tr}}} (\mathcal{R}^{e_i,e_j}(f; d^{e_j}) - \mathcal{R}^{e_j,e_i}(f; d^{e_i}))^2 + \beta \sum_{e \in \mathcal{E}_{tr}} \mathbb{E}_{p_e(X)} \left[ -\log d^e(\Phi(\mathbf{x})) \right]. \quad (9)$$

As in equation 6, the loss is comprised of the ERM and WRI penalties, with an additional log penalty term to regularize the learned representations. For full implementation details, refer to Appendix D.

## 4    Experiments

We evaluate our WRI implementation on synthetic and real-world datasets with distribution shifts, particularly focusing on cases of covariate shift in the invariant features. In all of the datasets, we select our model based on a validation set from the test environment, and we report the test set accuracy averaged over 5 random seeds with standard errors. Training validation selection assumes that the training and test data are drawn from similar distributions, which is not the case we aim to be robust to (Gulrajani & Lopez-Paz, 2020); test validation selection is more useful for demonstrating OOD capabilities (Ruan et al., 2021).

Our major baselines are ERM (Vapnik, 1991), IRM (Arjovsky et al., 2019), and VREx (Krueger et al., 2021). IRM and VREx are two other causally-motivated works that also search for conditional invariance as a signature of an underlying causal structure. Because WRI shares a similar theoretical grounding, we find it particularly important to compare our empirical performance with these works. Appendix E includes comparisons with other non-causal baselines, as well as additional experiments and details.

### 4.1    ColoredMNIST

ColoredMNIST (CMNIST) is a dataset proposed by Arjovsky et al. (2019) as a binary classification extension of MNIST. In this dataset, digit shapes are the invariant features and digit colors are the spurious features. The digit colors are more closely correlated to the labels than the digit shapes are, but the correlation between color and label is reversed in the test environment. The design of the dataset allows invariant predictors, which base their predictions on the digit shape, to outperform predictors that use spurious color information. We create a heteroskedastic variant on this dataset, HCMNIST, where we vary the label flip probability with the digit. We also create HCMNIST-CS, as a version of HCMNIST with invariant covariate shift, by enforcing different distributions of digits in each environment. For a predictor to perform well on the test set of these datasets, it must learn to predict based on the invariant features, even under heteroskedastic noise (present in both HCMNIST and HCMNIST-CS) and invariant covariate shift (HCMNIST-CS). Finally, we create ideal versions of these datasets where we simplify the image data to two-dimensional features of digit value and color. (More details on how these datasets were generated can be found in Appendix E.3.)

**Table 1:** Ideal HCMNIST digit and color penalty comparison (×1*e-3*).

| Algorithm | Dataset | Digit | Color | Min |
|---|---|---|---|---|
| WRI | HCMNIST | 0.0 | 4.7 | Digit ✓ |
| | HCMNIST-CS | 0.0 | 0.8 | Digit ✓ |
| VREx | HCMNIST | 0.0 | 5.4 | Digit ✓ |
| | HCMNIST-CS | 1.1 | 0.4 | Color ✗ |

**Table 2:** Accuracy on HCMNIST without covariate shift and with covariate shift

| Algorithm | HCMNIST | HCMNIST-CS | Avg |
|---|---|---|---|
| ERM | 53.8 ± 2.3 | 51.1 ± 0.9 | 52.4 |
| IRM | 67.2 ± 2.7 | 61.3 ± 4.7 | 64.3 |
| VREx | 67.4 ± 2.3 | 58.2 ± 2.3 | 62.8 |
| WRI | **75.1 ± 2.9** | **74.7 ± 3.8** | **74.9** |

**Table 3:** CMNIST OOD detection

| Algorithm | tpr@fpr=20% | tpr@fpr=40% | tpr@fpr=60% | tpr@fpr=80% | AUROC |
|---|---|---|---|---|---|
| ERM | 25.4 ± 3.6 | 45.6 ± 2.8 | 64.7 ± 1.7 | 82.2 ± 1.2 | 0.539 ± 0.018 |
| IRM | 29.8 ± 3.2 | 46.8 ± 2.8 | 66.4 ± 1.9 | 84.8 ± 0.9 | 0.565 ± 0.017 |
| VREx | 25.3 ± 2.5 | 43.5 ± 2.3 | 61.6 ± 1.5 | 80.8 ± 1.1 | 0.525 ± 0.014 |
| WRI | **36.7 ± 6.7** | **54.1 ± 5.8** | **68.1 ± 4.5** | **85.0 ± 1.5** | **0.595 ± 0.039** |

We use the ideal datasets to evaluate the WRI and VREx penalties with a predictor that uses only the digit value (i.e. an invariant predictor) and a predictor that uses only the color (i.e. a spurious predictor), and we report the results in Table 1. As anticipated, WRI registers zero penalty for the invariant predictor in both the HCMNIST and HCMNIST-CS scenarios. In contrast, VREx registers zero penalty for the invariant predictor on HCMNIST but a non-zero penalty on HCMNIST-CS, favoring the spurious predictor under heteroskedasticity and invariant covariate shift.

We compare our method with all baselines on the image (non-ideal) data in Table 2. These results demonstrate that WRI consistently performs well in both the heteroskedastic and the heteroskedastic and distribution shift settings, while VREx and IRM show significant degradation under invariant covariate shift.

**OOD detection performance of our learned densities** To test the estimated invariant density from WRI, we compare its utility as an OOD detector to the model confidences from ERM, IRM, and VREx. The evaluation is conducted on a modified CMNIST test split with mirrored digits. Since the estimated invariant density should be a function of the shape, we expect the invalid digits to have lower density estimates. We compute AUROC values for each experiment and report the results in Table 3. We observe that at all computed false positive rates, the true positive rate is higher for our method; this means that our learned density values are better for detecting OOD digits than the prediction confidences from other methods. For more details, see Appendix E.4.

## 4.2 Real-world datasets from DomainBed

We evaluate our method on 5 real-world datasets that are part of the DomainBed suite, namely VLCS (Fang et al., 2013), PACS (Li et al., 2017), OfficeHome (Venkateswara et al., 2017), TerraIncognita (Beery et al., 2018), and DomainNet (Peng et al., 2019). We run on ERM-trained features for computational efficiency, as these should still contain spurious information as well as the invariant information necessary to generalize OOD (Rosenfeld et al., 2022). In order to achieve a fair comparison, we evaluate all methods starting from the same set of pretrained features. We report the average performance across environments on each dataset in Table 4.

While the WRI predictor achieves higher accuracy than the baselines, we find that the performances are all similar. Methods like VREx that rely on assumptions like homoskedasticity may still see high accuracy on datasets that fulfill those assumptions. We note that, in addition to generalization accuracy, our method has the additional benefit of reporting when predictions are based on invariant features that have low probability in training. Additional information and results on DomainBed are provided in Appendix E.5.

**Table 4:** DomainBed results on feature data

| Algorithm | VLCS | PACS | OfficeHome | TerraIncognita | DomainNet | Avg |
|---|---|---|---|---|---|---|
| ERM | $76.5 \pm 0.2$ | $84.7 \pm 0.1$ | $64.5 \pm 0.1$ | $51.2 \pm 0.2$ | $33.5 \pm 0.1$ | 62.0 |
| IRM | $76.7 \pm 0.3$ | $84.7 \pm 0.3$ | $63.8 \pm 0.6$ | $52.8 \pm 0.3$ | $22.7 \pm 2.8$ | 60.1 |
| VREx | $76.7 \pm 0.2$ | $84.8 \pm 0.2$ | $64.6 \pm 0.2$ | $52.2 \pm 0.3$ | $26.6 \pm 2.1$ | 61.0 |
| WRI | $77.0 \pm 0.1$ | $85.2 \pm 0.1$ | $64.5 \pm 0.2$ | $52.7 \pm 0.3$ | $32.8 \pm 0.0$ | 62.5 |

## 5 Related works

**Invariant learning**  Much of the causal motivation for invariant learning stems from Schölkopf et al. (2012), who connected the invariant mechanism assumption to invariance in conditional distributions across multiple environments. Since then, invariant learning works have found that the invariant mechanism assumption results in additional forms of invariance, including invariance in the optimal predictor Arjovsky et al. (2019) and risk invariance under homoskedasticity (Krueger et al., 2021). Uncovering the underlying invariant mechanisms through explicit causal discovery (Peters et al., 2016) can be difficult to scale up to complex, high-dimensional data, so a significant branch of invariant learning instead leverages these additional forms of invariance to recover a predictor based on invariant mechanisms. While these methods do not provide all the guarantees typically offered by methods in causal discovery, they are faster and simpler in comparison, and can still provide theoretical guarantees for certain causal structures.

Learning theory provides another framework for understanding invariant learning. The seminal work of Ben-David et al. (2010) shows that the risk on a test environment is upper bounded by the error on the training environment(s), the total variation between the marginal distributions of training and test, and the difference in labeling functions between training and test. While the first term is minimized in ERM, the second term motivates learning marginally invariant features $\Psi(X)$ such that $p_e(\Psi(X^e))$ is invariant across $e$ (Pan et al., 2010; Baktashmotlagh et al., 2013; Ganin et al., 2016; Tzeng et al., 2017; Long et al., 2018; Zhao et al., 2018), and the third term motivates learning conditionally invariant features $\Phi(X)$ such that $p_e(Y^e|\Phi(X^e))$ is invariant across $e$ (Muandet et al., 2013; Koyama & Yamaguchi, 2020). Observing the importance of both approaches, several works even attempt to learn features that are both marginally and conditionally invariant (Zhang et al., 2013; Long et al., 2015; Li et al., 2021; Guo et al., 2021). Our work follows a similar vein: we focus on learning conditionally invariant features, under the assumption that these align with the underlying causal features, but we also reweight the conditionally invariant features to mitigate any shift in their marginal distributions.

**Importance weighting methods**  Starting from the early and influential method of importance weighting (Shimodaira, 2000), many methods for dealing with covariate shift use some form of weighting approach (Martin et al., 2023). Weighting methods work by weighting the loss with a density ratio of test features to training features (Shimodaira, 2000; Huang et al., 2006), or vice versa (Liu & Ziebart, 2014). However, these methods assume that the support of the features in the numerator is contained in the support of the features in the denominator. Further, they can perform poorly when the density ratio is large (i.e. in areas where the features in the denominator are much more rare); these large values lead to high variance and low effective sample size in the density ratio estimator (see Appendix B). For this reason, we propose density weighting rather than density ratio weighting, as density weighting is not limited by the support of any feature distribution and leads to better performance in practice, as established by Martin et al. (2023). Alternatively, we can also overcome the density ratio support limitation with a generalized form of importance weighting (Fang et al., 2024).

**Density estimates for downstream tasks**  Density estimates are useful for a variety of downstream tasks. They serve as an intuitive measure of epistemic uncertainty (Hüllermeier & Waegeman, 2021): in our work, the invariant feature density estimates approximate how much reliable (spurious-free) evidence we have in the training data. As a result, our density estimates are useful for detecting when samples are OOD, along the line of previous/concurrent work (Breunig et al., 2000; Charpentier et al., 2020; Peng et al., 2024). In general, density estimates are useful for building trustworthy systems, and they can be leveraged for both

fairness (Cho et al., 2020; Balunović et al., 2021) and calibration (Kuleshov & Deshpande, 2022; Deshpande & Kuleshov, 2023; Xiong et al., 2023).

## 6 Discussion

**Limitations of our method**   The invariant feature weighting functions in WRI allow its solutions to be robust to the case of heteroskedasticity and invariant covariate shift. However, these weighting functions are constrained by the invariant feature densities of each environment; we do not have access to these densities in practice, and can only recover an estimate of that density through alternating minimization. Future work could focus on better understanding the optimization process (in the style of Chen et al. (2022; 2024)), with the goal of recovering the invariant feature densities with more rigorous guarantees. We believe that an accurate invariant feature density estimate would be useful for reporting the confidence of predictions during deployment. In general, the question of how to better quantify the uncertainties of domain adaptation/generalization models remains an open line of research.

**Limitations of invariant prediction**   Invariant prediction methods assume there exists some generalizable relationship across training environments, but there are cases where it is impossible to extract the relationship. Notably for linear classification, Ahuja et al. (2021) proves that generalization is only possible if the support of invariant features in the test environment is a subset of the union of the supports in the training environments. For this reason, it is important to report when we are given data points that are rare in the invariant feature distribution; then, users know when a model is extrapolating outside of the training domain.

**Broader impacts**   Compared to other invariant learning methods, our WRI method offers the additional benefit of providing density estimates that allow a level of OOD detection, which can be useful in safety-critical situations. In general, we believe these estimates provide a useful measure of epistemic uncertainty that is not usually present in domain generalization methods.

## 7 Conclusion

Our work demonstrates the utility of weighted risk invariance for achieving invariant learning, even in the difficult case of heteroskedasticity and invariant covariate shift. We proved that weighted risk invariance is a signature of spurious-free prediction, and we further proved that enforcing weighted risk invariance allows us to recover a spurious-free predictor under a general causal setting. We proposed an efficient and useful method of weighting by the invariant feature densities, WRI, and we introduced an algorithm to practically enforce WRI by simultaneously learning the model parameters and the invariant feature densities. In our experiments, we demonstrated that the WRI predictor outperforms popular baselines under invariant covariate shift. Finally, we showed that our learned invariant feature density reports when invariant features are outside of the training distribution, a useful signal for the predictor's trustworthiness.

## 8 Reproducibility Statement

The code for generating the figures and empirical results can be found at `https://github.com/ginawong/weighted_risk_invariance/`.

## Acknowledgements

GW and RC were partially supported by a MURI from the Army Research Office under Grant No. W911NF-17-1-0304. This is part of the collaboration between US DOD, UK MOD, and UK Engineering and Physical Research Council (EPSRC) under the Multidisciplinary University Research Initiative. AL was partially supported by JHU Discovery Award, Amazon-JHU AI2AI Award, and a seed grant from JHU Institute of Assured Autonomy.

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

## A Proofs

**Proposition 1.** *Let Assumption 1 hold over a set of environments $\mathcal{E} \subseteq \mathcal{P}$. If a predictor $f$ is spurious-free over $\mathcal{E}$, then for every pair of environments $e_i, e_j \in \mathcal{E}$, their weighted risks are equal if their respective weighting functions $\alpha$ and $\beta$ obey $\alpha(\mathbf{x}_{inv})p_{e_i}(\mathbf{x}_{inv}) = \beta(\mathbf{x}_{inv})p_{e_j}(\mathbf{x}_{inv})$.*

*Proof.* The weighted risk for environment 1 is given by

$$\int_{\mathcal{Y}} \int_{\mathcal{X}_{\text{inv}}} \int_{\mathcal{X}_{\text{spu}}} p_1(\mathbf{x}_{\text{inv}}, \mathbf{x}_{\text{spu}}, y) \cdot \ell(f(\mathbf{x}_{\text{inv}}, \mathbf{x}_{\text{spu}}), y) \cdot \alpha(\mathbf{x}) \cdot d\mathbf{x}_{\text{spu}} \; d\mathbf{x}_{\text{inv}} \; dy.$$

If $f$ and $\alpha$ are invariant, then we can transform this integral as follows.

$$\int_{\mathcal{Y}} \int_{\mathcal{X}_{\text{inv}}} \int_{\mathcal{X}_{\text{spu}}} p_1(\mathbf{x}_{\text{inv}}, \mathbf{x}_{\text{spu}}, y) \cdot \ell(f(\mathbf{x}_{\text{inv}}, \mathbf{x}_{\text{spu}}), y) \cdot \alpha(\mathbf{x}_{\text{inv}}) \cdot d\mathbf{x}_{\text{spu}} \; d\mathbf{x}_{\text{inv}} \; dy$$

$$= \int_{\mathcal{Y}} \int_{\mathcal{X}_{\text{inv}}} \ell(f(\mathbf{x}_{\text{inv}}), y) \cdot \alpha(\mathbf{x}_{\text{inv}}) \int_{\mathcal{X}_{\text{spu}}} p_1(\mathbf{x}_{\text{inv}}, \mathbf{x}_{\text{spu}}, y) \cdot d\mathbf{x}_{\text{spu}} \; d\mathbf{x}_{\text{inv}} \; dy$$

$$= \int_{\mathcal{Y}} \int_{\mathcal{X}_{\text{inv}}} \ell(f(\mathbf{x}_{\text{inv}}), y) \cdot \alpha(\mathbf{x}_{\text{inv}}) \cdot p_1(\mathbf{x}_{\text{inv}}, y) \; d\mathbf{x}_{\text{inv}} \; dy$$

$$= \int_{\mathcal{Y}} \int_{\mathcal{X}_{\text{inv}}} \ell(f(\mathbf{x}_{\text{inv}}), y) \cdot \alpha(\mathbf{x}_{\text{inv}}) \cdot p_1(y | \mathbf{x}_{\text{inv}}) \cdot p_1(\mathbf{x}_{\text{inv}}) \; d\mathbf{x}_{\text{inv}} \; dy. \tag{A.1}$$

Similarly, if $\beta$ is invariant, then the weighted risk for environment 2 is given by

$$\int_{\mathcal{Y}} \int_{\mathcal{X}_{\text{inv}}} \ell(f(\mathbf{x}_{\text{inv}}), y) \cdot \beta(\mathbf{x}_{\text{inv}}) \cdot p_2(y | \mathbf{x}_{\text{inv}}) \cdot p_2(\mathbf{x}_{\text{inv}}) \cdot d\mathbf{x}_{\text{inv}} \; dy. \tag{A.2}$$

Because $p_e(y | \mathbf{x}_{\text{inv}})$ is the same for all $e$, then equation A.1 and equation A.2 are equal if $\alpha(\mathbf{x}_{\text{inv}})p_1(\mathbf{x}_{\text{inv}}) = \beta(\mathbf{x}_{\text{inv}})p_2(\mathbf{x}_{\text{inv}})$. $\qquad\square$

To show that weighted risk invariance leads to OOD generalization, we must first make an assumption on the diversity of our training environments. We give a definition for general position in a similar style to previous works (Arjovsky et al., 2019; Wald et al., 2021).

**Definition A.1.** We are given a set of $k$ environments such that $\binom{k}{2} > 2d_{\text{spu}}$, where $d_{\text{spu}}$ is the dimension of spurious features. Each environment $i$ has a weighted covariance $\Sigma^{i,j}$ and a weighted correlation $\mu^{i,j}$ with another environment $j$. We define these as

$$\Sigma^{i,j} = \mathbb{E}_{p_i(X^i, Y^i)} \left[ \mathbf{x} \mathbf{x}^{\mathsf{T}} \alpha_{ij}(\mathbf{x}_{\text{inv}}) \right]$$
$$\mu^{i,j} = 2 \cdot \mathbb{E}_{p_i(X^i, Y^i)} \left[ \mathbf{x} \, y \alpha_{ij}(\mathbf{x}_{\text{inv}}) \right].$$

We say that this set of environments is in general position if for any scalar $\gamma \in \mathbb{R}$ and all nonzero $\mathbf{x} \in \mathbb{R}^{d_{\text{spu}}}$,

$$\dim(\text{span}(\{(\Sigma^{i,j} - \Sigma^{j,i}) \mathbf{x} + \gamma(\mu^{i,j} - \mu^{j,i})\}_{i,j \in [k]})) = d_{\text{spu}}. \tag{A.3}$$

With this definition, we proceed to show that for a set of environments in general position, a linear regression model that satisfies weighted risk invariance must be spurious-free.

**Theorem 1.** *Consider a regression problem following the data generating process of equation 5. Let $\mathcal{E}$ be a set of environments with $|\mathcal{E}| > d_{spu}$ that satisfies general position. A linear regression model $f(\mathbf{x}) = \mathbf{w}^{\mathsf{T}} \mathbf{x}$ with $\mathbf{w} = [\mathbf{w}_{inv}, \mathbf{w}_{spu}]$ that satisfies weighted risk invariance w.r.t the squared loss must also satisfy $\mathbf{w}_{spu} = 0$. For density ratio weighting, general position holds with probability 1.*

*Proof.* Assume we have weighted risk invariance between environments 1 and 2, so

$$\int_{\mathcal{Y}} \int_{\mathcal{X}_{\text{inv}}} \int_{\mathcal{X}_{\text{spu}}} (\mathbf{w}^{\mathsf{T}}[\mathbf{x}_{\text{inv}}, \mathbf{x}_{\text{spu}}] - y)^2 \cdot p_1(\mathbf{x}_{\text{inv}}, \mathbf{x}_{\text{spu}}, y) \cdot \alpha(\mathbf{x}_{\text{inv}}) \cdot d\,\mathbf{x}_{\text{spu}} \ d\,\mathbf{x}_{\text{inv}} \ dy$$

$$= \int_{\mathcal{Y}} \int_{\mathcal{X}_{\text{inv}}} \int_{\mathcal{X}_{\text{spu}}} (\mathbf{w}^{\mathsf{T}}[\mathbf{x}_{\text{inv}}, \mathbf{x}_{\text{spu}}] - y)^2 \cdot p_2(\mathbf{x}_{\text{inv}}, \mathbf{x}_{\text{spu}}, y) \cdot \beta(\mathbf{x}_{\text{inv}}) \cdot d\,\mathbf{x}_{\text{spu}} \ d\,\mathbf{x}_{\text{inv}} \ dy. \tag{A.4}$$

When we expand the square, the left hand side becomes

$$\mathbf{w}^{\mathsf{T}} \left( \int_{\mathcal{Y}} \int_{\mathcal{X}_{\text{inv}}} \int_{\mathcal{X}_{\text{spu}}} [\mathbf{x}_{\text{inv}}, \mathbf{x}_{\text{spu}}][\mathbf{x}_{\text{inv}}, \mathbf{x}_{\text{spu}}]^{\mathsf{T}} \cdot p_1(\mathbf{x}_{\text{inv}}, \mathbf{x}_{\text{spu}}, y) \cdot \alpha(\mathbf{x}_{\text{inv}}) \cdot d\,\mathbf{x}_{\text{spu}} \ d\,\mathbf{x}_{\text{inv}} \ dy \right) \mathbf{w}$$

$$- 2\,\mathbf{w}^{\mathsf{T}} \int_{\mathcal{Y}} \int_{\mathcal{X}_{\text{inv}}} \int_{\mathcal{X}_{\text{spu}}} [\mathbf{x}_{\text{inv}}, \mathbf{x}_{\text{spu}}] y \cdot p_1(\mathbf{x}_{\text{inv}}, \mathbf{x}_{\text{spu}}, y) \cdot \alpha(\mathbf{x}_{\text{inv}}) \cdot d\,\mathbf{x}_{\text{spu}} \ d\,\mathbf{x}_{\text{inv}} \ dy$$

$$+ \int_{\mathcal{Y}} \int_{\mathcal{X}_{\text{inv}}} \int_{\mathcal{X}_{\text{spu}}} y^2 \cdot p_1(\mathbf{x}_{\text{inv}}, \mathbf{x}_{\text{spu}}, y) \cdot \alpha(\mathbf{x}_{\text{inv}}) \cdot d\,\mathbf{x}_{\text{spu}} \ d\,\mathbf{x}_{\text{inv}} \ dy.$$

We define

$$\Sigma^{1,2} = \int_{\mathcal{Y}} \int_{\mathcal{X}_{\text{inv}}} \int_{\mathcal{X}_{\text{spu}}} [\mathbf{x}_{\text{inv}}, \mathbf{x}_{\text{spu}}][\mathbf{x}_{\text{inv}}, \mathbf{x}_{\text{spu}}]^{\mathsf{T}} \cdot p_1(\mathbf{x}_{\text{inv}}, \mathbf{x}_{\text{spu}}, y) \cdot \alpha(\mathbf{x}_{\text{inv}}) \cdot d\,\mathbf{x}_{\text{spu}} \ d\,\mathbf{x}_{\text{inv}} \ dy$$

$$= \int_{\mathcal{X}_{\text{inv}}} \int_{\mathcal{X}_{\text{spu}}} [\mathbf{x}_{\text{inv}}, \mathbf{x}_{\text{spu}}][\mathbf{x}_{\text{inv}}, \mathbf{x}_{\text{spu}}]^{\mathsf{T}} \cdot p_1(\mathbf{x}_{\text{inv}}, \mathbf{x}_{\text{spu}}) \cdot \alpha(\mathbf{x}_{\text{inv}}) \cdot d\,\mathbf{x}_{\text{spu}} \ d\,\mathbf{x}_{\text{inv}}$$

$$\mu^{1,2} = 2 \int_{\mathcal{Y}} \int_{\mathcal{X}_{\text{inv}}} \int_{\mathcal{X}_{\text{spu}}} [\mathbf{x}_{\text{inv}}, \mathbf{x}_{\text{spu}}] y \cdot p_1(\mathbf{x}_{\text{inv}}, \mathbf{x}_{\text{spu}}, y) \cdot \alpha(\mathbf{x}_{\text{inv}}) \cdot d\,\mathbf{x}_{\text{spu}} \ d\,\mathbf{x}_{\text{inv}} \ dy$$

$$c^{1,2} = \int_{\mathcal{Y}} \int_{\mathcal{X}_{\text{inv}}} \int_{\mathcal{X}_{\text{spu}}} y^2 \cdot p_1(\mathbf{x}_{\text{inv}}, \mathbf{x}_{\text{spu}}, y) \cdot \alpha(\mathbf{x}_{\text{inv}}) \cdot d\,\mathbf{x}_{\text{spu}} \ d\,\mathbf{x}_{\text{inv}} \ dy$$

$$= \int_{\mathcal{Y}} \int_{\mathcal{X}_{\text{inv}}} y^2 \cdot p_1(\mathbf{x}_{\text{inv}}, y) \cdot \alpha(\mathbf{x}_{\text{inv}}) \cdot d\,\mathbf{x}_{\text{inv}} \ dy,$$

with analogous definitions for $\Sigma^{2,1}$, $\mu^{2,1}$, and $c^{2,1}$. Then equation A.4 can be written more succinctly as

$$\mathbf{w}^{\mathsf{T}} \Sigma^{1,2} \mathbf{w} - \mathbf{w}^{\mathsf{T}} \mu^{1,2} + c^{1,2} = \mathbf{w}^{\mathsf{T}} \Sigma^{2,1} \mathbf{w} - \mathbf{w}^{\mathsf{T}} \mu^{2,1} + c^{2,1}. \tag{A.5}$$

We decompose the quadratic term $\Sigma^{1,2}$ into invariant, spurious, and mixed components:

$$\Sigma^{1,2} = \int_{\mathcal{X}_{\text{inv}}} \int_{\mathcal{X}_{\text{spu}}} [\mathbf{x}_{\text{inv}}, \mathbf{x}_{\text{spu}}][\mathbf{x}_{\text{inv}}, \mathbf{x}_{\text{spu}}]^{\mathsf{T}} \cdot p_1(\mathbf{x}_{\text{inv}}, \mathbf{x}_{\text{spu}}) \cdot \alpha(\mathbf{x}_{\text{inv}}) \cdot d\,\mathbf{x}_{\text{spu}} \ d\,\mathbf{x}_{\text{inv}}$$

$$= \int_{\mathcal{X}_{\text{inv}}} \int_{\mathcal{X}_{\text{spu}}} \begin{bmatrix} \mathbf{x}_{\text{inv}} \mathbf{x}_{\text{inv}}^{\mathsf{T}} & \mathbf{x}_{\text{inv}} \mathbf{x}_{\text{spu}}^{\mathsf{T}} \\ \mathbf{x}_{\text{spu}} \mathbf{x}_{\text{inv}}^{\mathsf{T}} & \mathbf{x}_{\text{spu}} \mathbf{x}_{\text{spu}}^{\mathsf{T}} \end{bmatrix} \cdot p_1(\mathbf{x}_{\text{inv}}, \mathbf{x}_{\text{spu}}) \cdot \alpha(\mathbf{x}_{\text{inv}}) \cdot d\,\mathbf{x}_{\text{spu}} \ d\,\mathbf{x}_{\text{inv}}$$

$$= \int_{\mathcal{X}_{\text{inv}}} \begin{bmatrix} \mathbf{x}_{\text{inv}} \mathbf{x}_{\text{inv}}^{\mathsf{T}} & \mathbf{0} \\ \mathbf{0} & \mathbf{0} \end{bmatrix} \cdot p_1(\mathbf{x}_{\text{inv}}) \cdot \alpha(\mathbf{x}_{\text{inv}}) \cdot d\,\mathbf{x}_{\text{inv}}$$

$$+ \int_{\mathcal{X}_{\text{inv}}} \int_{\mathcal{X}_{\text{spu}}} \begin{bmatrix} \mathbf{0} & \mathbf{0} \\ \mathbf{0} & \mathbf{x}_{\text{spu}} \mathbf{x}_{\text{spu}}^{\mathsf{T}} \end{bmatrix} \cdot p_1(\mathbf{x}_{\text{inv}}, \mathbf{x}_{\text{spu}}) \cdot \alpha(\mathbf{x}_{\text{inv}}) \cdot d\,\mathbf{x}_{\text{spu}} \ d\,\mathbf{x}_{\text{inv}}.$$

$$+ \int_{\mathcal{X}_{\text{inv}}} \int_{\mathcal{X}_{\text{spu}}} \begin{bmatrix} \mathbf{0} & \mathbf{x}_{\text{inv}} \mathbf{x}_{\text{spu}}^{\mathsf{T}} \\ \mathbf{x}_{\text{spu}} \mathbf{x}_{\text{inv}}^{\mathsf{T}} & \mathbf{0} \end{bmatrix} \cdot p_1(\mathbf{x}_{\text{inv}}, \mathbf{x}_{\text{spu}}) \cdot \alpha(\mathbf{x}_{\text{inv}}) \cdot d\,\mathbf{x}_{\text{spu}} \ d\,\mathbf{x}_{\text{inv}}.$$

$$= \begin{bmatrix} \Sigma^{1,2}_{\text{inv}} & \mathbf{0} \\ \mathbf{0} & \mathbf{0} \end{bmatrix} + \begin{bmatrix} \mathbf{0} & \mathbf{0} \\ \mathbf{0} & \Sigma^{1,2}_{\text{spu}} \end{bmatrix} + \Sigma^{1,2}_{mix}.$$

We break down the third term $\Sigma_{mix}^{1,2}$ in more detail, following the data-generating process for $\mathbf{x}_{\text{spu}}$.

$$\Sigma_{mix}^{1,2} = \int_{\mathcal{X}_{\text{inv}}} \int_{\mathcal{X}_{\text{spu}}} \begin{bmatrix} \mathbf{0} & \mathbf{x}_{\text{inv}}\, \mathbf{x}_{\text{spu}}^{\mathsf{T}} \\ \mathbf{x}_{\text{spu}}\, \mathbf{x}_{\text{inv}}^{\mathsf{T}} & \mathbf{0} \end{bmatrix} \cdot p_1(\mathbf{x}_{\text{inv}}, \mathbf{x}_{\text{spu}}) \cdot \alpha(\mathbf{x}_{\text{inv}}) \cdot d\,\mathbf{x}_{\text{spu}}\ d\,\mathbf{x}_{\text{inv}}$$

$$= \int_{\mathcal{X}_{\text{inv}}} \int_{\varepsilon} \int_{\eta} \begin{bmatrix} \mathbf{0} & \mathbf{x}_{\text{inv}}\, \mu_i^{\mathsf{T}}\, \mathbf{x}_{\text{inv}}^{\mathsf{T}}\, \mathbf{w}_{\text{inv}}^* + \mathbf{x}_{\text{inv}}\, \mu_i^{\mathsf{T}} \varepsilon^{\mathsf{T}} + \mathbf{x}_{\text{inv}}\, \eta^{\mathsf{T}} \\ (\mathbf{w}_{\text{inv}}^*)^{\mathsf{T}}\, \mathbf{x}_{\text{inv}}\, \mu_i\, \mathbf{x}_{\text{inv}}^{\mathsf{T}} + \varepsilon \mu_i\, \mathbf{x}_{\text{inv}}^{\mathsf{T}} + \eta\, \mathbf{x}_{\text{inv}}^{\mathsf{T}} & \mathbf{0} \end{bmatrix} \cdots$$

$$\cdots p_1(\varepsilon) \cdot p_1(\eta) \cdot p_1(\mathbf{x}_{\text{inv}}) \cdot \alpha(\mathbf{x}_{\text{inv}}) \cdot d\eta\ d\varepsilon\ d\,\mathbf{x}_{\text{inv}}$$

$$= \int_{\mathcal{X}_{\text{inv}}} \begin{bmatrix} \mathbf{0} & \mathbf{x}_{\text{inv}}\, \mu_i^{\mathsf{T}}\, \mathbf{x}_{\text{inv}}^{\mathsf{T}}\, \mathbf{w}_{\text{inv}}^* \\ (\mathbf{w}_{\text{inv}}^*)^{\mathsf{T}}\, \mathbf{x}_{\text{inv}}\, \mu_i\, \mathbf{x}_{\text{inv}}^{\mathsf{T}} & \mathbf{0} \end{bmatrix} \cdot p_1(\mathbf{x}_{\text{inv}}) \cdot \alpha(\mathbf{x}_{\text{inv}}) \cdot d\,\mathbf{x}_{\text{inv}}$$

$$+ \int_{\mathcal{X}_{\text{inv}}} \int_{\varepsilon} \begin{bmatrix} \mathbf{0} & \mathbf{x}_{\text{inv}}\, \mu_i^{\mathsf{T}} \\ \mu_i\, \mathbf{x}_{\text{inv}}^{\mathsf{T}} & \mathbf{0} \end{bmatrix} \cdot \varepsilon \cdot p_1(\varepsilon) \cdot p_1(\mathbf{x}_{\text{inv}}) \cdot \alpha(\mathbf{x}_{\text{inv}}) \cdot d\varepsilon\ d\,\mathbf{x}_{\text{inv}}$$

$$+ \int_{\mathcal{X}_{\text{inv}}} \int_{\eta} \begin{bmatrix} \mathbf{0} & \mathbf{x}_{\text{inv}}\, \eta^{\mathsf{T}} \\ \eta\, \mathbf{x}_{\text{inv}}^{\mathsf{T}} & \mathbf{0} \end{bmatrix} \cdot p_1(\eta) \cdot p_1(\mathbf{x}_{\text{inv}}) \cdot \alpha(\mathbf{x}_{\text{inv}}) \cdot d\eta\ d\,\mathbf{x}_{\text{inv}}\,.$$

Both $\varepsilon$ and $\eta$ are mean-zero random variables that are independent of $\mathbf{x}_{\text{inv}}$. This means the second and third terms go to zero, leaving us with only the first term. We further decompose the first term, now explicitly labeling the dimensions of the 0 submatrices for clarity.

$$\Sigma_{mix}^{1,2} = \begin{bmatrix} I_{d_{\text{inv}}} & \mathbf{0}_{d_{\text{inv}} \times 1} \\ \mathbf{0}_{d_{\text{spu}} \times d_{\text{inv}}} & \mu_i \end{bmatrix} \Sigma_{w-inv}^{1,2} \begin{bmatrix} I_{d_{\text{inv}}} & \mathbf{0}_{d_{\text{inv}} \times d_{\text{spu}}} \\ \mathbf{0}_{1 \times d_{\text{inv}}} & \mu_i^{\mathsf{T}} \end{bmatrix},$$

where

$$\Sigma_{w-inv}^{1,2} = \int_{\mathcal{X}_{\text{inv}}} \begin{bmatrix} \mathbf{0}_{d_{\text{inv}} \times d_{\text{inv}}} & \mathbf{x}_{\text{inv}}(\mathbf{w}_{\text{inv}}^*)^{\mathsf{T}}\, \mathbf{x}_{\text{inv}} \\ \mathbf{x}_{\text{inv}}^{\mathsf{T}}\, \mathbf{w}_{\text{inv}}^*\, \mathbf{x}_{\text{inv}}^{\mathsf{T}} & \mathbf{0}_{1 \times 1} \end{bmatrix} \cdot p_1(\mathbf{x}_{\text{inv}}) \cdot \alpha(\mathbf{x}_{\text{inv}}) \cdot d\,\mathbf{x}_{\text{inv}}$$

$$= \begin{bmatrix} \mathbf{0}_{d_{\text{inv}} \times d_{\text{inv}}} & \mathbf{v}^{1,2} \\ (\mathbf{v}^{1,2})^{\mathsf{T}} & \mathbf{0}_{1 \times 1} \end{bmatrix},$$

with $\mathbf{v}^{1,2} = \mathbf{x}_{\text{inv}}(\mathbf{w}_{\text{inv}}^*)^{\mathsf{T}}\, \mathbf{x}_{\text{inv}}$ being a $d_{\text{inv}}$-dimensional column vector.

We also separate the linear term $\mu^{1,2}$ into invariant and spurious components:

$$\mu^{1,2} = 2 \int_{\mathcal{Y}} \int_{\mathcal{X}_{\text{inv}}} \int_{\mathcal{X}_{\text{spu}}} [\mathbf{x}_{\text{inv}}, \mathbf{x}_{\text{spu}}] y \cdot p_1(\mathbf{x}_{\text{inv}}, \mathbf{x}_{\text{spu}}, y) \cdot \alpha(\mathbf{x}_{\text{inv}}) \cdot d\,\mathbf{x}_{\text{spu}}\ d\,\mathbf{x}_{\text{inv}}\ dy$$

$$= 2 \int_{\mathcal{Y}} \int_{\mathcal{X}_{\text{inv}}} [\mathbf{x}_{\text{inv}}, \mathbf{0}] y \int_{\mathcal{X}_{\text{spu}}} p_1(\mathbf{x}_{\text{inv}}, \mathbf{x}_{\text{spu}}, y) \cdot \alpha(\mathbf{x}_{\text{inv}}) \cdot d\,\mathbf{x}_{\text{spu}}\ d\,\mathbf{x}_{\text{inv}}\ dy$$

$$+ 2 \int_{\mathcal{Y}} \int_{\mathcal{X}_{\text{spu}}} [\mathbf{0}, \mathbf{x}_{\text{spu}}] y \int_{\mathcal{X}_{\text{inv}}} p_1(\mathbf{x}_{\text{inv}}, \mathbf{x}_{\text{spu}}, y) \cdot \alpha(\mathbf{x}_{\text{inv}}) \cdot d\,\mathbf{x}_{\text{inv}}\ d\,\mathbf{x}_{\text{spu}}\ dy$$

$$= 2 \int_{\mathcal{Y}} \int_{\mathcal{X}_{\text{inv}}} [\mathbf{x}_{\text{inv}}, \mathbf{0}] y \cdot p_1(y|\,\mathbf{x}_{\text{inv}}) \cdot p_1(\mathbf{x}_{\text{inv}}) \cdot \alpha(\mathbf{x}_{\text{inv}}) \cdot d\,\mathbf{x}_{\text{inv}}\ dy$$

$$+ 2 \int_{\mathcal{Y}} \int_{\mathcal{X}_{\text{spu}}} [\mathbf{0}, \mathbf{x}_{\text{spu}}] y \int_{\mathcal{X}_{\text{inv}}} p_1(\mathbf{x}_{\text{inv}}, \mathbf{x}_{\text{spu}}, y) \cdot \alpha(\mathbf{x}_{\text{inv}}) \cdot d\,\mathbf{x}_{\text{inv}}\ d\,\mathbf{x}_{\text{spu}}\ dy$$

$$= [\mu_{\text{inv}}^{1,2}, \mathbf{0}] + [\mathbf{0}, \mu_{\text{spu}}^{1,2}].$$

Finally, we break down the constant term $c^{1,2}$.

$$c^{1,2} = \int_{\mathcal{Y}} \int_{\mathcal{X}_{\text{inv}}} y^2 \cdot p_1(\mathbf{x}_{\text{inv}}, y) \cdot \alpha(\mathbf{x}_{\text{inv}}) \cdot d\,\mathbf{x}_{\text{inv}}\ dy$$

$$= \int_{\mathcal{Y}} \int_{\mathcal{X}} y^2 \cdot p_1(y|\,\mathbf{x}_{\text{inv}}) \cdot p_1(\mathbf{x}_{\text{inv}}) \cdot \alpha(\mathbf{x}_{\text{inv}}) \cdot d\,\mathbf{x}_{\text{inv}}\ dy.$$

Given the condition on the weighting functions $\alpha(\mathbf{x}_{\text{inv}})p_1(\mathbf{x}_{\text{inv}}) = \beta(\mathbf{x}_{\text{inv}})p_2(\mathbf{x}_{\text{inv}})$, it is clear that $\Sigma_{\text{inv}}^{1,2} = \Sigma_{\text{inv}}^{2,1}$, $\mathbf{v}^{1,2} = \mathbf{v}^{2,1}$, $\mu_{\text{inv}}^{1,2} = \mu_{\text{inv}}^{2,1}$, and $c^{1,2} = c^{2,1}$. With this information, we define $\gamma := 2\,\mathbf{w}_{\text{inv}}^{\mathsf{T}}\mathbf{v}^{1,2} - 1$ and simplify equation A.5 to

$$\mathbf{w}_{\text{spu}}^{\mathsf{T}}\,\Sigma_{\text{spu}}^{1,2}\,\mathbf{w}_{\text{spu}} + \gamma(\mu_{\text{spu}}^{1,2})^{\mathsf{T}}\,\mathbf{w}_{\text{spu}} = \mathbf{w}_{\text{spu}}^{\mathsf{T}}\,\Sigma_{\text{spu}}^{2,1}\,\mathbf{w}_{\text{spu}} + \gamma(\mu_{\text{spu}}^{2,1})^{\mathsf{T}}\,\mathbf{w}_{\text{spu}}. \tag{A.6}$$

Since equation A.6 holds for each pairwise combination of environments, then

$$\mathbf{w}_{\text{spu}}^{\mathsf{T}}(\Sigma_{\text{spu}}^{i,j} - \Sigma_{\text{spu}}^{j,i})\,\mathbf{w}_{\text{spu}} + \gamma(\mu_{\text{spu}}^{i,j} - \mu_{\text{spu}}^{j,i})^{\mathsf{T}}\,\mathbf{w}_{\text{spu}} = 0 \quad \forall i,j \in [k]. \tag{A.7}$$

We define the $\binom{k}{2} \times d_{\text{spu}}$ matrix

$$M = \begin{bmatrix} \mathbf{w}_{\text{spu}}^{\mathsf{T}}(\Sigma_{\text{spu}}^{1,2} - \Sigma_{\text{spu}}^{2,1}) + \gamma(\mu_{\text{spu}}^{1,2} - \mu_{\text{spu}}^{2,1})^{\mathsf{T}} \\ \mathbf{w}_{\text{spu}}^{\mathsf{T}}(\Sigma_{\text{spu}}^{1,3} - \Sigma_{\text{spu}}^{3,1}) + \gamma(\mu_{\text{spu}}^{1,3} - \mu_{\text{spu}}^{3,1})^{\mathsf{T}} \\ \vdots \\ \mathbf{w}_{\text{spu}}^{\mathsf{T}}(\Sigma_{\text{spu}}^{k,k-1} - \Sigma_{\text{spu}}^{k-1,k}) + \gamma(\mu_{\text{spu}}^{k,k-1} - \mu_{\text{spu}}^{k-1,k})^{\mathsf{T}} \end{bmatrix}.$$

The environments are in general position, so matrix $M$ is full rank for any nonzero $\mathbf{w}_{\text{spu}}$. That means that there is no nonzero vector $\mathbf{x}$ that solves

$$M\,\mathbf{x} = \mathbf{0}. \tag{A.8}$$

If there is no nonzero solution to equation A.8, then there is no nonzero $\mathbf{w}_{\text{spu}}$ that solves equation A.7. Thus, equation A.5 implies that $\mathbf{w}_{\text{spu}} = 0$. $\qquad \square$

We now examine the case of density ratio weighting. For a pair of environments $i'$ and $j'$ with the weighting constraint $\alpha(\mathbf{x}_{\text{inv}})p_{e_{i'}}(\mathbf{x}_{\text{inv}}) = \beta(\mathbf{x}_{\text{inv}})p_{e_{j'}}(\mathbf{x}_{\text{inv}})$, we let $\alpha(\mathbf{x}_{\text{inv}}) = 1$ so that $\beta(\mathbf{x}_{\text{inv}}) = p_{e_{i'}}(\mathbf{x}_{\text{inv}})/p_{e_{j'}}(\mathbf{x}_{\text{inv}})$. The weighting function $\alpha$ no longer depends on an environment-specific distribution; instead, it is the same for all pairs of environments that include environment $i'$. This weighting allows us to give an additional result based on a smaller set of equations centered around environment $i'$.

Under density ratio weighting, covariance $\Sigma^{1,i}$ is identical for all environment indices $i$, as is mean $\mu^{1,i}$. This means that if we generalize equation A.6 to all environment pairs with environment 1, the left hand side of the equation is identical. Then for some scalar $t \in \mathbb{R}$,

$$\mathbf{w}_{\text{spu}}^{\mathsf{T}}\,\Sigma_{\text{spu}}^{i,1}\,\mathbf{w}_{\text{spu}} + \gamma(\mu_{\text{spu}}^{i,1})^{\mathsf{T}}\,\mathbf{w}_{\text{spu}} = t \quad \forall i \in [k] \setminus 1. \tag{A.9}$$

This system of equations can be written as a $k \times d_{\text{spu}}$ matrix

$$M' = \begin{bmatrix} \mathbf{w}_{\text{spu}}^{\mathsf{T}}\,\Sigma_{\text{spu}}^{1,1} + \gamma(\mu_{\text{spu}}^{1,1})^{\mathsf{T}} & 1 \\ \mathbf{w}_{\text{spu}}^{\mathsf{T}}\,\Sigma_{\text{spu}}^{2,1} + \gamma(\mu_{\text{spu}}^{2,1})^{\mathsf{T}} & 1 \\ \vdots & \\ \mathbf{w}_{\text{spu}}^{\mathsf{T}}\,\Sigma_{\text{spu}}^{k,1} + \gamma(\mu_{\text{spu}}^{k,1})^{\mathsf{T}} & 1 \end{bmatrix}.$$

The steps by which the density ratio weighting simplifies matrix $M$ to $M'$ should illustrate how our general position definition can also be stated more simply under this weighting, in line with the definitions given by similar works. Then, a straightforward application of Theorem 10 of Arjovsky et al. (2019) or Lemma S4 of Wald et al. (2021) gives our final result: that the subset of environments in our setting that are outside of general position has measure zero.

# B  On the sample complexity of different weighting functions

Our work differs from other works on weighting to mitigate distribution shift. We are the first to propose weighting by functions of the invariant features; this allows us to use weighted risks to recover a spurious-free predictor under shift, whereas most forms of weighting only focus on mitigating shift, without any invariant learning mechanisms. Still, work on the generalization bounds of importance weighting (to mitigate shift) can be readily extended to our case. In this section, we use the arguments of Cortes et al. (2010) to show that convergence guarantees are improved by selecting weights with a lower variance and lower maximum value. This result explains some of the benefits of using density weights over density ratio weights.

We fix $f \in H$, where $H$ denotes the hypothesis set under consideration. For some weighting functions $\alpha$ and $\beta$ such that $\alpha(\mathbf{x})p_{e_1}(\mathbf{x}) = \beta(\mathbf{x})p_{e_2}(\mathbf{x})$, we define the weighted risk over two environments $e_1$ and $e_2$ to be

$$\mathcal{R}(f) = \mathbb{E}_{p_{e_1}(X,Y)}\left[\alpha(\mathbf{x})\ell(f(\mathbf{x}),y)\right] = \mathbb{E}_{p_{e_2}(X,Y)}\left[\beta(\mathbf{x})\ell(f(\mathbf{x}),y)\right]. \tag{B.10}$$

The empirical weighted risk is defined to be

$$\hat{\mathcal{R}}_\alpha(f) = \frac{1}{|\mathcal{D}_{e_1}|} \sum_{\mathbf{x},y \in \mathcal{D}_{e_1}} \alpha(\mathbf{x})\ell(f(\mathbf{x}),y), \text{ and}$$

$$\hat{\mathcal{R}}_\beta(f) = \frac{1}{|\mathcal{D}_{e_2}|} \sum_{\mathbf{x},y \in \mathcal{D}_{e_2}} \beta(\mathbf{x})\ell(f(\mathbf{x}),y). \tag{B.11}$$

Following the proof of Theorem 1 from Cortes et al. (2010), we assume that $\ell(f(\mathbf{x}),y) \in [0,1]$. we define a random variable $Z_\alpha = \alpha(X)\ell(f(X),Y) - \mathcal{R}(f)$ where $X,Y$ are drawn from $p_{e_1}$. Then, $\sigma^2(Z_\alpha) = \mathbb{E}_{p_{e_1}(X,Y)}\left[\alpha^2(\mathbf{x})\ell^2(f(\mathbf{x}),y)\right] - \mathcal{R}(f)^2$. Applying Bernstein's inequality yields

$$\Pr[\mathcal{R}(f) - \hat{\mathcal{R}}_\alpha(f) > \varepsilon] \leq \exp\left(\frac{-|\mathcal{D}_{e_1}|\varepsilon^2}{2\sigma^2(Z) + 2\varepsilon M_\alpha/3}\right), \tag{B.12}$$

where $M_\alpha = \max\{1, \sup_{\mathbf{x}} \alpha(\mathbf{x})\}$. Setting $\delta$ equal to the bound from Eq. (B.12) and solving for $\varepsilon$, it follows that with probability at least $1 - \delta$ the following bound holds for any $\delta > 0$:

$$\left|\mathcal{R}(f) - \hat{\mathcal{R}}_\alpha(f)\right| \leq \frac{M_\alpha \log \frac{1}{\delta}}{3|\mathcal{D}_{e_1}|} + \sqrt{\frac{M_\alpha^2 \log^2 \frac{1}{\delta}}{9|\mathcal{D}_{e_1}|^2} + \frac{2\sigma^2(Z_\alpha) \log \frac{1}{\delta}}{|\mathcal{D}_{e_1}|}}$$

$$\leq \frac{2M_\alpha \log \frac{1}{\delta}}{3|\mathcal{D}_{e_1}|} + \sqrt{\frac{2\sigma^2(Z_\alpha) \log \frac{1}{\delta}}{|\mathcal{D}_{e_1}|}}. \tag{B.13}$$

Following the same steps, we get a bound for $\left|\mathcal{R}(f) - \hat{\mathcal{R}}_\beta(f)\right|$. If we combine these events with the union bound, then the probability of both of these bounds holding is at least $1 - 2\delta$.

Note that for density ratio weights, the maximum values $M_\alpha$ and $M_\beta$ can be large (e.g. when a sample is much more rare in one environment than another); this also leads to the variances $\sigma^2(Z_\alpha)$ and $\sigma^2(Z_\beta)$ being large. In contrast, density weights tend to have a lower maximum value and lower variance.

We emphasize that this section does not provide a sample complexity bound for our method; it should be treated as a exploratory comparison of different weighting functions only, with the caveat that our result relies on several assumptions that would not hold in practice. We fix the hypothesis $f$, but it would typically be learned from training data. We also assume the optimal weighting functions are known, when these would need to be learned from the data/trained model as well.

## C   Optimization of WRI for the 2D Gaussian case

In the main body of the paper, we show that for a set of invariant weighting functions, weighted risk invariance is a necessary and sufficient condition for spurious-free prediction. This demonstrates the importance of correcting for invariant covariate shift, and extends the importance weighting method to the invariant learning setting. However, we encounter a chicken-and-egg problem in the implementation of our method; specifically, that we weight by the density of the invariant features, without having access to the invariant features in practice. At best, we only have access to the density of the joint features, both spurious and invariant.

In this section, we show that for a 2D case of our regression setting in equation 5, optimizing for WRI leads us to learn a model that discards the spurious features, *even when we start with the density of the joint features.* In particular, we see from Figure C.1 that optimizing for risk weighted by the joint feature density eventually leads to the weights on the spurious features going to zero, leaving only the weights on the invariant features.

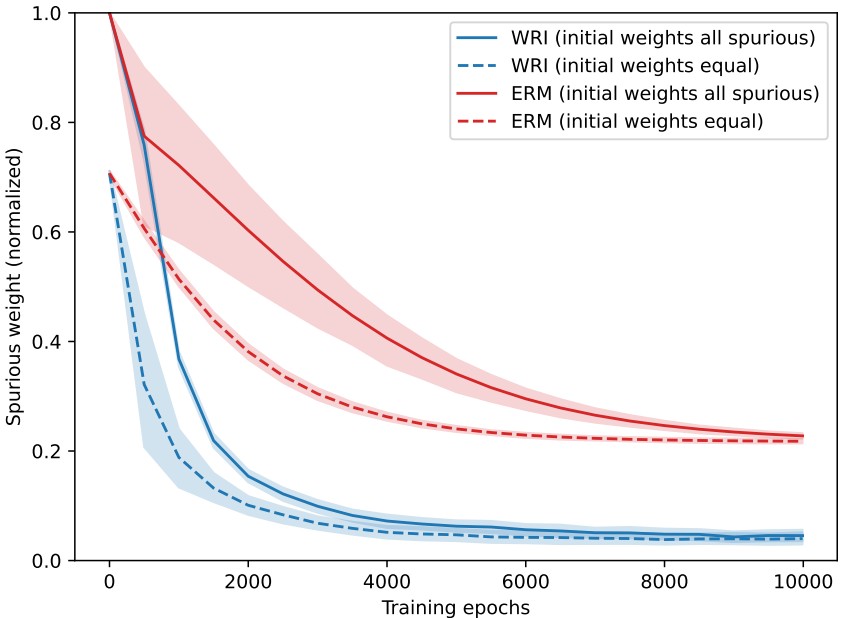

**Figure C.1:** Given data of the form $[\mathbf{x}_{\mathrm{inv}}, \mathbf{x}_{\mathrm{spu}}]$ following our setting in equation 5, we train a model with a two-parameter feature layer, where one weight scales the invariant features and one weight scales the spurious features. For both the WRI and ERM methods, we do two initializations where we initialize the model with equal feature weights and with weights on the spurious features only. We find that optimizing WRI always leads to the spurious weights going to zero, whereas optimizing ERM converges to nonzero spurious weights.

### C.1   Experiment details

The data generation process for this experiment follows equation 5, where each data point $\mathbf{x} = [\mathbf{x}_{\mathrm{inv}}, \mathbf{x}_{\mathrm{spu}}]$. This data is then used to train a two-layer linear model, where the feature layer has two weight parameters: one weight for the invariant features, and one for the spurious features. The model layers do not have any activation functions, so they are straightforward linear transformations.

To decouple our investigation from the difficulties associated with density estimation, we assume that $X_{\mathrm{inv}}$ is Gaussian-distributed. In this way, we have a closed form solution for the density of the features. When we optimize for WRI, we therefore weight by the exact feature density.

We use this setup to optimize for both the WRI and ERM methods. To start, we initialize the model with either equal feature weights (where the invariant and spurious feature weights are equal), or with the

zero invariant feature weight (so all the weight is on the spurious features). We find that, regardless of initialization, the WRI-trained model learns to reject the spurious weight. This means that as the model trains, we learn to weight by an invariant feature density and predict from the invariant features only. The ERM-trained model, on the other hand, never reaches a nonzero spurious weight.

## C.2 Discussion

The results of Figure C.1 are promising evidence that invariant feature reweighting can be realized in a practical implementation, even though we only have access to the input data and not the invariant features. However, please note that we only show this for a specific case of our regression setting, where the invariant and spurious features are easily separable and the invariant features $X_{\text{inv}}$ follow a Gaussian distribution. If we were to run this experiment on data from an unknown distribution, the results would be less clear: our weighting functions would depend on how correctly we can estimate feature density, an inexact operation in high dimensional cases.

---

**Algorithm D.1:** WRI with model-based density

---

**Parameters:**
    $n$: total number of optimization steps
    $\omega$: density update frequency
    $n_d$: number of density update steps

**initialize** random model weights for $f$ and $d^e$ for all $e \in \mathcal{E}_{tr}$
**initialize** Adam (Kingma & Ba, 2017) optimizer for $f$
**initialize** Adam optimizer for $d^e$
**for** $i = 1 \ldots n$ **do**
    sample featurized minibatch from all environments
    compute $\mathcal{L}$ equation 9 on minibatch and step $f$ optimizer
    **if** $i - 1$ is a multiple of $\omega$ **then**
        **repeat** $n_d$ **times**
            sample featurized minibatch from all environments
            compute $\mathcal{L}$ on minibatch and step $d^e$ optimizer

---

# D   Implementation details

When the optimal invariant predictor was first introduced with IRM, it was first motivated as a constrained optimization problem that searched over all spurious-free predictors for the predictor with the lowest training loss. The practical objective IRMv1 was then introduced, where the constraint was approximated by a gradient penalty. Many follow-up works took a similar path, where a penalty is used as some form of invariance constraint. Our penalty is weighted risk invariance; according to our theory, models with zero weights on spurious features will satisfy the WRI constraint. Therefore, a learning rule that finds the most accurate classifier satisfying WRI will learn an optimal hypothesis that relies on the invariant features alone.

We implement this learning rule with alternating minimization; our specific algorithm is described in Algorithm D.1, and a graphical representation of our network architecture is shown in Figure D.2. This is a complex optimization procedure, and we do not derive guarantees on its convergence to invariant representations in this work. We simply motivate this implementation with the knowledge that a spurious-free predictor would minimize the WRI penalty, which we optimize for when the penalty coefficient in equation 9 is sufficiently large. We test that optimizing WRI on the observable joint feature density still focuses model weights on the invariant features in Appendix C. In the main body of the paper, we empirically show that our objective is effective at recovering a generalizable predictor, as well as density estimates with a meaningful signal for OOD detection. Later in this section (Appendix D.3), we also test the quality of our density estimates for a 2D case of our regression setting in equation 5.

## D.1   Optimizing WRI leads to invariant features

To provide additional intuition on why optimizing for WRI leads to learning invariant features, let us consider trying to reweight a representation $\Phi(X)$ that is not invariant. Specifically, assume that for two environments, $p_1(Y \mid \Phi(X)) \neq p_2(Y \mid \Phi(X))$. In what follows, we demonstrate why WRI would reject such a representation. We focus on density ratio reweighting for simplicity, so that $\tilde{p} = p_1(\Phi(X)) \cdot p_2(Y \mid \Phi(X)))$.

To satisfy invariance under this reweighting, we must have $\tilde{p}(Y \mid \Phi(X)) = p_1(Y \mid \Phi(X))$. Clearly this does not hold, since $\tilde{p}(Y \mid \Phi(X)) = p_2(Y \mid \Phi(X))$, and this lack of invariance can be identified from data under mild assumptions on overlap. Therefore, enforcing this type of weighted invariance constraint (i.e. matching of the conditionals $p_1(Y \mid \Phi(X))$ and the density ratio weighted distribution $\tilde{p}(Y \mid \Phi(X)))$) would rule out a non-invariant representation $\Phi(X)$. Conversely, an invariant representation would naturally satisfy this constraint.

The gap between this constraint and weighted risk invariance lies in the fact that we only enforce invariance of the risk, not the full conditionals (we do this because imposing risk invariance is more tractable in practice).

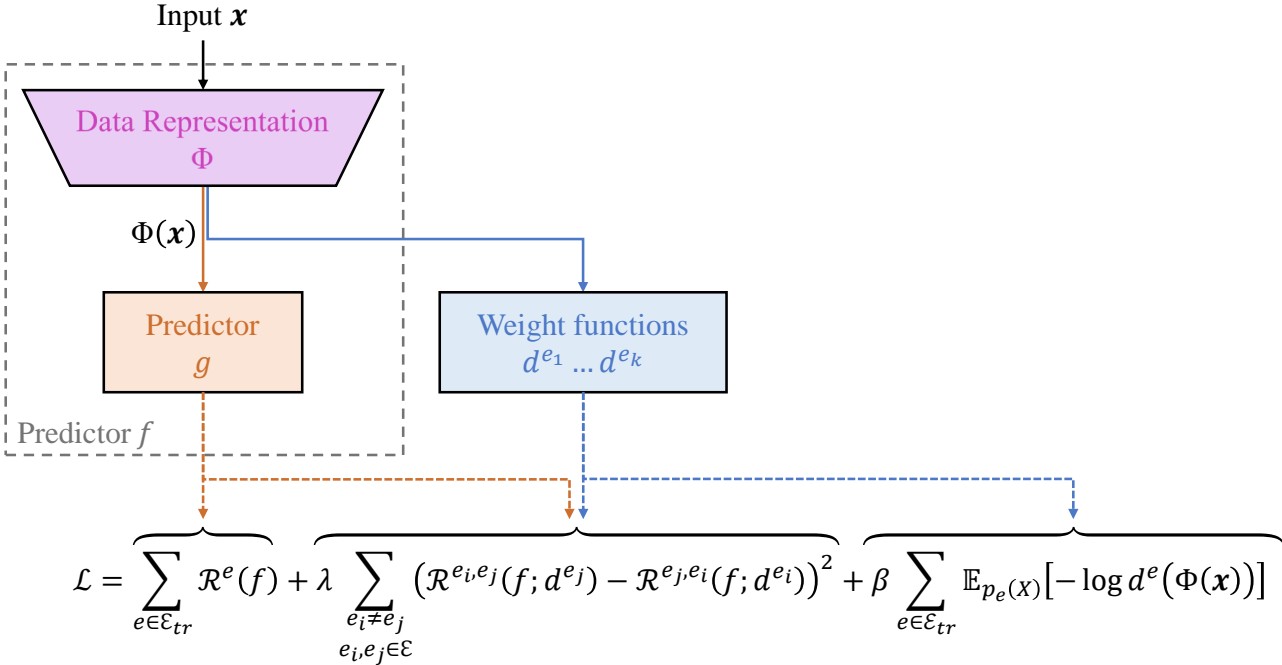

**Figure D.2:** Graphic overview of our network architecture. The red lines indicate back-propagation paths for components that impact the first two terms. The blue lines indicate the paths for the last two terms.

While the two are not equivalent, there are cases where they coincide. For example, Theorems 1 and 2 and the experiments of Krueger et al. (2021) support the claim that risk invariance learns an invariant predictor.

A subtle but important point is that these theorems presuppose the existence of a risk invariant predictor. Under the invariant covariate shift we consider, some invariant features must be discarded to obtain a risk invariant predictor (e.g. in the 2D Gaussian case, a risk invariant predictor would have to discard the causal feature completely). With weighted risk invariance, this issue is alleviated. By using (for example) density ratio reweighting, the optimal invariant predictor becomes risk invariant again.

### D.2 Practical strategies

We observe that the magnitude of the WRI regularization term from equation 9 varies significantly in practice, making it difficult to choose a good value for $\lambda$. This is true even when the true invariant density is known. We employ two strategies to deal with this. First, we constrain the density model to predict values within a pre-defined range by applying a sigmoid activation on the final prediction with constant scale and shift factor. Additionally, we divide the WRI term by the average negative log-likelihood, which also helps to decouple the empirical risk from the WRI regularization term.

We allow different optimization parameters for the prediction model and density estimation models. Both optimizers have a different learning rate, weight decay, batch size, and $\lambda$ penalty. (The range of hyperparameter values we use are provided with the DomainBed details in Appendix E.5.)

### D.3 Density estimation quality

Optimizing WRI with exact feature density values successfully recovers an invariant predictor, as we show in Appendix C. Yet, density values are often estimated with some inaccuracy in practice. In this section, we investigate the accuracy of our alternating minimization method in estimating feature densities. We use the Gaussian case of our regression setting in equation 5, as detailed in Appendix C. Combining this with a

linear model for feature learning provides a closed-form solution for the feature density that we can compare against our estimated densities at each alternating step

**Learning the parameters of a density model**  When we know the distribution family we are estimating the densities of, we should set up our density estimation model so that we are learning the parameters of the same distribution. In this case, the log penalty pulls us toward a maximum likelihood estimate of the parameters that would generate the observed feature densities.

**Learning an unconstrained density estimation model**  When we make no assumptions about the distribution family, we opt to use a multilayer perceptron (MLP) to model our unnormalized density models, with a sigmoid activation function to limit the minimum and maximum output. As we specify in equation 9, we fit our density models with the sum of the WRI penalty and a negative log penalty. The WRI penalty attempts to fulfill the weighted risk constraint (where learned weighting/functions $\alpha$ and $\beta$ satisfy $\alpha(\mathbf{x})p_{e_i}(\mathbf{x}_{inv}) = \beta(\mathbf{x})p_{e_j}(\mathbf{x}_{inv})$). Since the total mass is not fixed with this model, the log penalty primarily acts as a regularization to disincentivize low estimates.

After the density models are trained and the unnormalized density estimates computed, we scale the density estimates by a constant of proportionality $C$ to obtain *proportional density estimates*, where $C$ is the scalar that minimizes the mean squared error (MSE) between the estimated and exact densities. We then normalize the MSE between the proportional density estimates and the exact densities, so that an error of one corresponds to a density estimator that predicts a constant (which would reduce the WRI penalty to the VREx penalty). We show how the normalized/relative MSE of the proportional density estimates goes down as we train in Figure D.3.

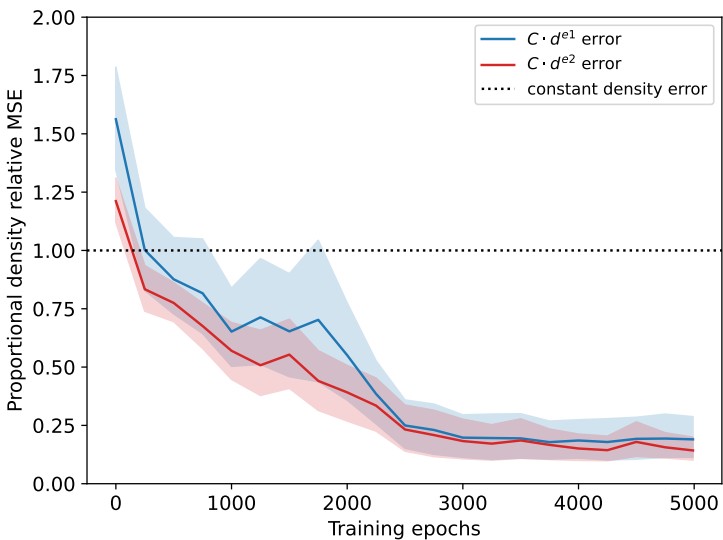

**Figure D.3:** Under alternating minimization with an unconstrained density estimation model, the relative MSE of the proportional density estimates goes down over time.

**Where are density estimates expected to be accurate?**  To understand how the accuracy of the density estimates is impacted by the support of the environments, consider a data model with two environments: $e_1$ and $e_2$. The density estimate $d^{e_2}$ is evaluated only on samples from environment $e_1$, and similarly the density estimate $d^{e_1}$ is evaluated only on samples from environment $e_2$. For this reason, we expect the WRI penalty to incentivize accurate density estimates for $d^{e_1}$ where the ground truth density $p_{e_2}$ is high, and for $d^{e_2}$ where $p_{e_1}$ is high. In regions where the density $p_{e_2}$ is small, we expect the learning of $d^{e_1}$ to be influenced by other factors, including the log penalty and the tendency toward learning smooth functions (due to other regularization, the continuous nature of commonly used activation functions, etc.) The combination of these factors results in the extrapolation of the density estimates outside the region of support overlap, as shown in Figure D.4. This extrapolation does not significantly impact the WRI penalty, as at least one probability density will be low in this region, so the penalty will be low as well.

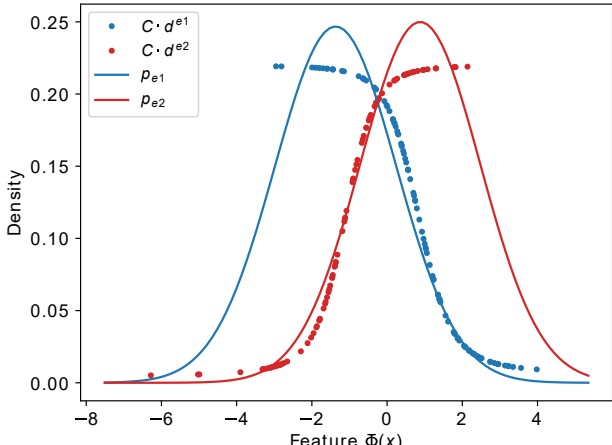

**Figure D.4:** We train an unconstrained density estimation model for 3000 epochs (to convergence) and compare the proportional density learned for each feature to the ground truth density of the invariant features. We see that for a given environment $e_i$, the learned density accurately approximates the shape of the ground truth density where it overlaps with the other environment $e_j$.

# E    Extensions on experiments and additional experimental details

## E.1    Toy dataset hyperparameters (Figure 1 details)

To produce Figure 1, we define a toy dataset consisting of the three environments described here. Environments 1 and 2 are used for training and environment 3 is used as test. We let $Y = \mathbb{1}[X_{\text{inv}} + \varepsilon > 0]$, where $\varepsilon$ is a standard normal random variable and $\mathbb{1}$ is the indicator function. The following data distributions are used:

$$X_{\text{inv}}^1 \sim N(0, 2^2), \qquad X_{\text{inv}}^2 \sim N(0, \left(\tfrac{1}{2}\right)^2), \qquad X_{\text{inv}}^3 \sim N(0, 3^2),$$
$$X_{\text{spu}}^1|Y = 0 \sim N(1, \left(\tfrac{1}{2}\right)^2), \qquad X_{\text{spu}}^2|Y = 0 \sim N(1, 2^2), \qquad X_{\text{spu}}^3|Y = 0 \sim N(-1, 1^2),$$
$$X_{\text{spu}}^1|Y = 1 \sim N(-1, \left(\tfrac{1}{2}\right)^2), \qquad X_{\text{spu}}^2|Y = 1 \sim N(-1, 2^2), \text{ and} \qquad X_{\text{spu}}^3|Y = 1 \sim N(1, 1^2).$$

We sample $10^4$ points from each environment. The predictor is a simple linear model. The ERM, IRM, VREx, and WRI objectives are optimized using scikit-learn (Pedregosa et al., 2011). We use a penalty weight of $10^5$ for IRM, a penalty weight of 1 for VREx, and a penalty weight of 1500 for WRI. We note that VREx converges to an absurd solution when using a larger weight, despite generally using a much larger weight in the DomainBed implementation. The penalty weights were roughly optimized by eye to achieve the best qualitative results (regarding invariance).

## E.2    Experiments on multi-dimensional synthetic dataset

**Data generated following structural equation model**    We construct a multi-class classification simulation based on the linear causal model shown in Figure 2. The spurious and invariant distributions are drawn from normal distributions $X_{\text{inv}} \sim N(\mu^e, \Sigma^e)$ and $X_{\text{spu}}^e|Y = y \sim N(\mu^{e,y}, \Sigma^{e,y})$ and $X$ is the concatenation of $X_{\text{inv}}$ and $X_{\text{spu}}$. To simulate the classification scenario, we define the class label to be $Y = \arg\max_y \mathbf{w}_y^\mathsf{T} \mathbf{x}_{\text{inv}} + \varepsilon_y$ where $\varepsilon_y \sim N(0, \sigma_y)$.

Rather than specifying all of the distribution parameters manually, we sample them randomly according to

$$\begin{aligned}
\mu^e &\sim N(0, I\sigma_{\text{inv}}^2), & \Sigma^e &= RandCov(1 - \Delta_{\text{inv}}, 1 + \Delta_{\text{inv}}), & \mathbf{w}_y &\sim U(\mathbb{S}_{d_{\text{inv}}}), \\
\mu^{e,y} &\sim N(0, I\sigma_{\text{spu}}^2), & \Sigma^{e,y} &= RandCov(\tfrac{1}{2}, \tfrac{3}{2}), & \sigma_y &= \sigma_y
\end{aligned} \tag{E.14}$$

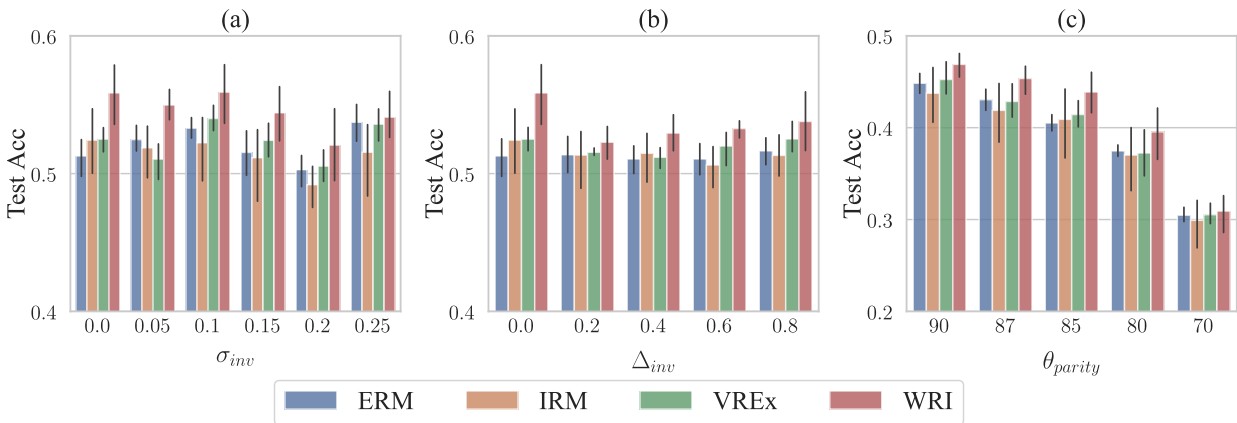

**Figure E.5:** Our simulated classification setting from §E.2 where we measure the test accuracy under different shifts in the invariant features. (a) We vary the average distance between invariant distributions. (b) We vary the covariance matrices between invariant distributions. (c) With $\sigma_{\mathrm{inv}} = 0.15$, $\Delta_{\mathrm{inv}} = 0.5$, we vary the correlation of spurious features between the training environments with test having the opposite correlation. In all cases, the WRI predictor outperforms other methods on average.

where $\mathbb{S}_{d_{\mathrm{inv}}}$ is the $d_{\mathrm{inv}}$-dimensional hypersphere, and $RandCov(a, b)$ is a covariance matrix-generating random variable that selects the square root of the eigenvalues i.i.d. from $U(a, b)$.

This formulation allows us to specify the data simulation using four scalar values: $\sigma_{\mathrm{inv}}$, $\sigma_{\mathrm{spu}}$, $\Delta_{\mathrm{inv}}$, and $\sigma_y$. The $\sigma_{\mathrm{inv}}$ and $\sigma_{\mathrm{spu}}$ terms define the variability of the means between different invariant and spurious distributions respectively. For example, when $\sigma_{\mathrm{inv}}$ is zero, $X_{\mathrm{inv}}$ will have the same mean value across all environments. $\Delta_{\mathrm{inv}}$ is a value between 0 and 1 that defines how much the invariant covariance matrices vary between environments, and $\sigma_y$ specifies the noise in the label generation process.

Figure E.5 shows how we compare to other baselines when we run on data with 20 dimensions ($d_{\mathrm{inv}} = 10$), with 4 training environments and 1 test. We see that (a) with isotropic covariance, we outperform other baselines when the invariant distribution means overlap and when we increase the distance between the means. Further, (b) with identical means, we outperform other baselines when the invariant distributions all have isotropic covariance and when the covariance matrices vary between distributions. Both cases represent the covariate shift in the invariant features, which our predictor is designed to be more robust to. On average, our improvement over the baselines is more significant when we have isotropic covariance for all environments and increase the distance between the environment means; we believe this is because the isotropic covariance allows for a clearer (more controlled) case of covariate shift.

**Testing dependence on controlled spurious correlation** We run a modified version of our simulation that tests the misleading nature of the spurious features. Similar to the ColoredMNIST approach of defining the train and test environments to have opposite color-label correlation, we randomly select a vector for each class and ensure that its correlation with the mean of the spurious distributions is positive for each training environment and negative for the test environment. This is accomplished by first sampling the data as in equation E.14, flipping correlation if necessary, then applying a transform to ensure that the means are in a hypercone with opening angle $\theta_{parity}$. As $\theta_{parity}$ decreases from 90°, the spurious data increasingly predicts $Y$ consistently across training environments, making it more challenging to disentangle the spurious and invariant features. In Figure E.5(c), we see that although decreasing $\theta_{parity}$ degrades the performance of all methods, WRI still performs better than the other baselines.

**Additional discussion of simulation hyperparameters** Here, we further discuss the hyperparameters $\sigma_{\mathrm{inv}}$, $\Delta_{\mathrm{inv}}$, $\sigma_y$, and $\sigma_{\mathrm{spu}}$, and how they impact the data generation.

$\sigma_{\mathrm{inv}}$ is a non-negative scalar that encapsulates the distance between invariant data for each environment. When this value is zero, the invariant distributions are all centered on the origin; when this value is large enough, the average distance between invariant distributions increases.

$\Delta_{\mathrm{inv}}$ is a scalar in $[0, 1]$ that encapsulates how much the covariance matrices of the invariant distributions differ between environments. When this value is zero, all environments have identity covariance. Intuitively, this value is similar to $\sigma_{\mathrm{inv}}$ in the sense that larger values will produce invariant distributions that overlap less between environments.

$\sigma_y$ is the only directly specified parameter of the data model and it controls the amount of label noise present. When set to zero, the label is a deterministic function of $X_{\mathrm{inv}}$; when set very high, $Y$ becomes more difficult to predict from the invariant data.

$\sigma_{\mathrm{spu}}$ is a non-negative scalar that encapsulates the distance between spurious data distributions. As this value gets larger, the spurious distributions are less likely to overlap both between classes and between environments. This has two primary effects. First, within a single environment, less overlap will make spurious data more predictive of the label. However, because the locations of the distributions change between environments, this also means that the spurious nature of $X_{\mathrm{spu}}$ should be more apparent during training.

One caveat to our method of data generation is that controlling for a uniform number of classes is difficult since the class label is a function of $X_{\mathrm{inv}}$. For this reason, we sample a new set of data model parameters when one class is overrepresented in any environment by a factor of 1.5 times the expected uniform representation (e.g. for 5 classes, we resample when more than 30% of data from any environment have the same label). This also ensures that any models with accuracy above 1.5 times uniform are making non-trivial predictions.

**Controlling the spurious correlation** For the experiment reported in Figure E.5 (c), we explicitly manipulate the data model to ensure a higher level of spurious correlation via the $\theta_{parity}$ hyperparameter. This produces a scenario similar to ColoredMNIST, where the correlation between the label and spurious data (color) is relatively consistent between training environments ($+90\%$ and $+80\%$) but has the opposite relationship in test ($-90\%$). This means that an algorithm that learns to rely on color will be heavily penalized at test time. In this section, we describe the mechanism behind $\theta_{parity}$ in more detail.

We begin by sampling distribution parameters for all the invariant and conditional spurious datasets (the same as in prior experiments). Next, we select a random unit vector $\mathbf{c}_y \sim U(\mathbb{S}_{d_{\mathrm{spu}}})$ for each class label. Our goal is to ensure that the spurious mean $\mu^{e,y}$ correlates positively with $\mathbf{c}_y$ for each of the training environments and negatively with the test environment. In other words, the training and test environments have opposite *parity* w.r.t. the hyperplane. We achieve this by negating $\mu^{e,y}$ when necessary, i.e.

$$\mu^{e,y} \leftarrow \begin{cases} \mathrm{sign}(\mathbf{c}_y^{\mathsf{T}} \mu^{e,y}) \mu^{e,y} & \text{if } e \in \mathcal{E}_{tr} \\ -\mathrm{sign}(\mathbf{c}_y^{\mathsf{T}} \mu^{e,y}) \mu^{e,y} & \text{otherwise} \end{cases}. \tag{E.15}$$

While this ensures the desired correlation, the relationship is not very strong due to the fact that random vectors in high-dimensional space tend to be nearly orthogonal. Therefore, we introduce $\theta_{parity}$.

For each class, we transform $\mu^{e,y}$ again to ensure it is inside the hypercone with axis $\mathbf{c}_y$ and opening angle $\theta_{parity}$. This is accomplished by rotating $\mu^{e,y}$ towards the axis so that the angle to the axis is scaled by a factor of $\theta_{parity}/90°$. The identity transform is, therefore, synonymous with $\theta_{parity} = 90°$, and as $\theta_{parity}$ decreases towards $0°$ the spurious data becomes more consistently predictive of the label.

**Numerical Results** For experiments on the synthetic datasets, we integrate the DomainBed implementations of IRM and VREx. All experiments use 20 dimensional data (10 invariant dimensions and 10 spurious dimensions). They also all use 5 environments (1 test and 4 training) and 5 class labels. The numerical values visualized in the Figure E.5 bar graphs are shown in Tables E.1, E.2, and E.3.

### E.3 Heteroskedastic CMNIST

ColoredMNIST (CMNIST) is a dataset proposed by Arjovsky et al. (2019) as a binary classification extension of MNIST where the shapes of the digits are invariant features and the colors of the digits are spurious features that are more tightly correlated to the label than the shapes are, with the correlation between the color and the label being reversed in the test environment. Specifically, digits 0–4 and 5–9 are classes 0 and 1

**Table E.1:** Sweep $\sigma_{\text{inv}}$ ($\Delta_{\text{inv}} = 0, \sigma_y = 0.5, \sigma_{\text{spu}} = 1$). Plotted in Figure E.5 (a)

| Algorithm | $\sigma_{\text{inv}}$ 0.0 | 0.05 | 0.1 | 0.15 | 0.2 | 0.25 | 0.3 |
|---|---|---|---|---|---|---|---|
| ERM | $51.3 \pm 1.6$ | $52.5 \pm 1.1$ | $53.3 \pm 0.9$ | $51.5 \pm 2.0$ | $50.3 \pm 1.3$ | $53.7 \pm 1.4$ | $54.8 \pm 2.1$ |
| IRM | $52.4 \pm 2.7$ | $51.9 \pm 2.3$ | $52.2 \pm 2.7$ | $51.2 \pm 3.2$ | $49.2 \pm 1.8$ | $51.5 \pm 3.2$ | $54.3 \pm 3.3$ |
| VREx | $52.5 \pm 1.0$ | $51.1 \pm 1.5$ | $54.0 \pm 1.0$ | $52.4 \pm 1.4$ | $50.5 \pm 1.3$ | $53.6 \pm 1.3$ | $56.8 \pm 1.1$ |
| WRI | $55.9 \pm 2.5$ | $55.0 \pm 1.3$ | $55.9 \pm 2.3$ | $54.4 \pm 2.3$ | $52.1 \pm 3.1$ | $54.1 \pm 1.8$ | $56.1 \pm 2.6$ |

**Table E.2:** Sweep $\Delta_{\text{inv}}$ ($\sigma_{\text{inv}} = 0, \sigma_y = 0.5, \sigma_{\text{spu}} = 1$). Plotted in Figure E.5 (b)

| Algorithm | $\Delta_{\text{inv}}$ 0.0 | 0.2 | 0.4 | 0.6 | 0.8 |
|---|---|---|---|---|---|
| ERM | $51.3 \pm 1.6$ | $51.4 \pm 1.5$ | $51.1 \pm 1.2$ | $51.1 \pm 1.3$ | $51.7 \pm 1.2$ |
| IRM | $52.4 \pm 2.7$ | $51.3 \pm 2.4$ | $51.5 \pm 2.1$ | $50.6 \pm 1.7$ | $51.3 \pm 1.8$ |
| VREx | $52.5 \pm 1.0$ | $51.6 \pm 0.3$ | $51.2 \pm 0.9$ | $52.0 \pm 1.5$ | $52.5 \pm 1.3$ |
| WRI | $55.9 \pm 2.5$ | $52.3 \pm 1.4$ | $52.9 \pm 1.6$ | $53.3 \pm 0.7$ | $53.8 \pm 2.5$ |

**Table E.3:** Sweep $\theta_{parity}$ ($\sigma_{\text{inv}} = 0.15, \Delta_{\text{inv}} = 0.5, \sigma_y = 0.5, \sigma_{\text{spu}} = 1$). Plotted in Figure E.5 (c)

| Algorithm | $\theta_{parity}$ 90° | 87° | 85° | 80° | 70° |
|---|---|---|---|---|---|
| ERM | $44.8 \pm 1.2$ | $43.0 \pm 1.4$ | $40.5 \pm 1.1$ | $37.5 \pm 0.7$ | $30.5 \pm 0.9$ |
| IRM | $43.8 \pm 3.6$ | $41.9 \pm 3.9$ | $40.9 \pm 4.2$ | $37.0 \pm 4.0$ | $29.9 \pm 3.0$ |
| VREx | $45.2 \pm 2.0$ | $42.9 \pm 1.9$ | $41.4 \pm 1.6$ | $37.2 \pm 2.7$ | $30.5 \pm 1.3$ |
| WRI | $46.9 \pm 1.5$ | $45.4 \pm 1.9$ | $43.9 \pm 2.6$ | $39.5 \pm 3.2$ | $30.9 \pm 2.2$ |

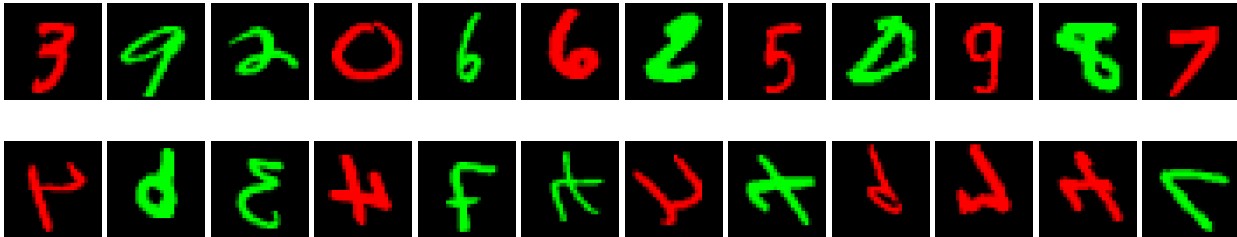

**Figure E.6:** Examples of CMNIST in-distribution (top) and out-of-distribution (bottom) used for OOD detection. The out-of-distibution samples are generated by flipping a subset of digits horizontally or vertically.

respectively; after injecting 25% label noise to all environments, the red digits in the first two environments have an 80% and 90% chance of belonging to class 0, while the red digits in the third environment have a 10% chance. The design of the dataset allows invariant predictors (that only base their predictions on the shape of the digits) to outperform predictors that use spurious color information, ideally achieving 75% accuracy on all environments.

**Heteroskedastic CMNIST with Covariate Shift**   Variants of CMNIST have been proposed, with the aim of incorporating additional forms of distribution shift (Ahuja et al., 2020; Wu et al., 2020; Krueger et al., 2021). To demonstrate the efficacy of our method under heteroskedastic covariate shift, we construct a heteroskedastic variant of CMNIST, which we call HCMNIST, where label flips occur for digits 0, 1, 5, and 6 with probability 5% and label flips occur for other digits with probability 25%. Additionally, we generate a variant of HCMNIST with covariate shift, HCMNIST-CS, where we redistribute data among environments. We place 65% of the digits 0, 1, 5, and 6 into the first training environment and 5% into the second environment (with the remaining 30% in test). The remaining digits are distributed so that all environments have the same number of samples.

The empirical results on the HCMNIST with and without covariate shift are shown in Table 2. These experiments use the practical version of the WRI algorithm and are computed using the same evaluation strategy as DomainBed. The results demonstrate that the practical WRI implementation performs well in both the heterogeneous case with and without covariate shift. Conversely, both IRM and VREx have significant degradation when covariate shift is introduced.

We also create an idealized version of these datasets that simplifies the images into two-dimensional features consisting of the digit value and the color. We evaluate the WRI penalty and VREx penalty using three optimal predictors: one that operates on only the digit value (invariant), one that operates on the color value (spurious), and one that operates on both. The results of this experiment are presented in Table 1. As expected, we find that both WRI and VREx have zero penalty on HCMNIST. However, in the presence of covariate shift, the VREx penalty for the invariant predictor is greater than both the penalty for the spurious classifier and the penalty for the mixed classifier. This demonstrates a simple and concrete case where VREx does not select for an invariant classifier, suggesting that the degradation in VREx performance under shift in Table 2 can be attributed to a true failure of VREx to recover the invariant predictor.

### E.4   Out-of-distribution Detection with CMNIST

To evaluate the out-of-distribution (OOD) detection performance on CMNIST, we first train a model on the two training environments of (traditional) CMNIST, then assess the model's ability to classify sample digits as in-domain or out-of-domain on a modified version of the CMNIST test environment. The modified test environment is a mix of unaltered digits for the "in-domain" samples and flipped digits for the "out-of-domain" samples. Specifically, we horizontally flip digits 3, 4, 7, and 9, and vertically flip digits 4, 6, 7, and 9. In this way, we create CMNIST samples that would not exist in the training distribution but still appear plausible. Figure E.6 provides examples of these in-domain and out-of-domain samples. Note that we only flip digits that are unlikely to be confused with real digits after flipping.

In order to measure OOD performance, we require an OOD score for each sample that predicts if the sample is in-distribution. Each OOD score is then compared against its in-distribution label to compute a Receiver Operating Characteristic (ROC) curve. This curve captures the trade-off between the true positive rate and false positive rate at various confidence thresholds. Lacking an explicit density estimate for ERM, IRM, and VREx, we instead use the maximum class prediction score for the OOD score. We then compare the effectiveness of WRI's density estimate as an OOD score against ERM, IRM, and VREx. Finally, we calculate the area under the ROC curve (AUROC) as an aggregate metric. The results, presented in Table 3, indicate that the WRI density estimates more accurately detect OOD digits than the prediction confidence scores from the other methods.

## E.5 DomainBed experimental details and results

### E.5.1 Featurizer

We use a ResNet50 model to train featurizers using the default ERM DomainBed parameters for each dataset. We train one featurizer corresponding to each test environment with both 32 and 64 dimensional features. Each featurizer is trained on the training environments for a fixed number of steps. Once trained, the features are pre-computed and all experiments are then performed on these features. The dimensionality of the features used is selected randomly as a hyperparameter. For each experiment, we use a single hidden-layer MLP with ReLU activation which operates directly on ResNet features.

### E.5.2 Hyperparameters

Since we are training on pre-extracted features, running for 5000 steps is unnecessary. Instead, we reduce the total number of steps per experiment to 500 and increase the default learning rate to 1e-2. Because we have fewer steps than the original (non-featurized) DomainBed implementation, we reduce the annealing iterations on IRM and VREx by a factor of 10 and limit the maximum annealing steps to 500. Otherwise, VREx and IRM would nearly always finish training before annealing has finished. Additionally, we define $\Phi$ to be an identity function, as we found this to produce the most consistent results.

We specified the following DomainBed hyperparameters and selected them according to the following distributions. Note that in DomainBed, the default hyperparameters are used during the first hyperparameter seed, and random hyperparameters are selected for subsequent seeds. For any hyperparameters not listed here, we use the defaults provided by DomainBed.

- Learning-rate - default: 1e-2, random: log-uniform over $[5e\text{-}3, 1e\text{-}1]$.

- IRM-anneal iters - default 50, random: log-uniform over $[1, 500]$. Number of steps before penalty weight is increased.

- VREx-anneal iters - default 50, random: log-uniform over $[1, 500]$. Number of steps before penalty weight is increased.

- Featurizer dimensions - default: 64, random: discrete uniform from $\{32, 64\}$. The dimensionality of the pretrained features used.

- WRI-$\lambda$ - default: 1, random: log-uniform over $[1e\text{-}1, 5e1]$. The penalty weight used when computing $\mathcal{L}$ in the predictor optimization step.

- WRI-annealing - default: 0, random: discrete uniform from $\{0, 10\}$. Number of steps before WRI regularization term is included in loss function.

- WRI-density update freq ($\omega$) - default: 1, random: discrete uniform from $\{1, 2, 4, 8\}$. Number of predictor optimization steps between density optimization steps.

- WRI-density learning rate - default: 2e-2, random: log-uniform over $[1e\text{-}2, 5e\text{-}2]$. Learning rate used for the density estimate optimizer.

- WRI-density weight decay - default: 1e-5, random: log-uniform over $[1e\text{-}6, 1e\text{-}2]$. The weight decay used for the density estimate optimizer.

- WRI-density batch size - default: 256, random: discrete uniform from $\{128, 256\}$. The batch size used when optimizing for density estimates.

- WRI-density $\lambda$ - default: 5, random: log-uniform from $[5, 50]$. The penalty weight used when computing $\mathcal{L}$ in the density optimization step.

- WRI-density $\beta$ - default: 2e2, random: log-uniform from $[5e\text{-}2, 5]$. The negative log penalty weight used when computing $\mathcal{L}$ in the density optimization step.

- WRI-density steps ($n_d$) - default: 4, random: discrete uniform from $\{4, 16, 32\}$. The number of optimization steps taken each time the density estimators are updated.

- WRI-min density - default: 0.05, random: uniform from $[0.01, 0.2]$. The minimum density imposed via scaled shifted sigmoid activation on density estimator models.

- WRI-max density - default: 1, random: uniform from $[0.4, 2]$. The maxmimum density imposed via scaled shifted sigmoid activation on density estimator models.

### E.5.3 DomainBed with additional baselines

We run DomainBed experiments on additional baselines to place our work in larger context, expanding Table 4 to Table E.4. Specifically, we also compare to GroupDRO (Sagawa et al., 2019), Mixup (Zhang et al., 2017), MLDG (Li et al., 2018), and CORAL (Sun & Saenko, 2016). These are the (current) top-performing methods that are implemented and tested in the DomainBed test suite that can be used with an ERM-trained featurizer—as that is sufficient to learn the invariant features necessary to train an OOD predictor (Rosenfeld et al., 2022), but also significantly speeds up training. We did not include methods that are not implemented in DomainBed, but it is worth mentioning examples like MIRO (Cha et al., 2022) that demonstrate state-of-the-art performance on image data. Note that all of the aforementioned methods are different lines of work; they are not causally motivated, but can sometimes have better generalization accuracy than methods with a causal basis.

**Table E.4:** DomainBed results on feature data, with additional non-causal baselines

| Algorithm | VLCS | PACS | OfficeHome | TerraIncognita | DomainNet | Avg |
|-----------|------|------|------------|----------------|-----------|-----|
| ERM | $76.5 \pm 0.2$ | $84.7 \pm 0.1$ | $64.5 \pm 0.1$ | $51.2 \pm 0.2$ | $33.5 \pm 0.1$ | 62.0 |
| IRM | $76.7 \pm 0.3$ | $84.7 \pm 0.3$ | $63.8 \pm 0.6$ | $52.8 \pm 0.3$ | $22.7 \pm 2.8$ | 60.1 |
| GroupDRO | $77.0 \pm 0.2$ | $84.8 \pm 0.1$ | $65.1 \pm 0.1$ | $51.3 \pm 0.2$ | $30.9 \pm 0.1$ | 61.9 |
| Mixup | $76.6 \pm 0.2$ | $85.0 \pm 0.1$ | $66.1 \pm 0.0$ | $52.3 \pm 0.7$ | $33.1 \pm 0.1$ | 62.6 |
| MLDG | $75.8 \pm 0.2$ | $82.3 \pm 0.2$ | $64.8 \pm 0.2$ | $48.3 \pm 0.2$ | $33.0 \pm 0.1$ | 60.8 |
| CORAL | $76.5 \pm 0.2$ | $85.1 \pm 0.2$ | $65.0 \pm 0.1$ | $51.8 \pm 0.4$ | $33.5 \pm 0.1$ | 62.5 |
| VREx | $76.7 \pm 0.2$ | $84.8 \pm 0.2$ | $64.6 \pm 0.2$ | $52.2 \pm 0.3$ | $26.6 \pm 2.1$ | 61.0 |
| WRI | $77.0 \pm 0.1$ | $85.2 \pm 0.1$ | $64.5 \pm 0.2$ | $52.7 \pm 0.3$ | $32.8 \pm 0.0$ | 62.5 |

### E.5.4 Individual dataset results

This section contains the individual DomainBed results on VLCS, PACS, OfficeHome, TerraIncognita, and DomainNet. The Average column from each dataset table is reported in Table E.4. For all other dataset tables, the column labels indicates which environment was held out for test.

**VLCS**

| Algorithm | C | L | S | V | Avg |
|-----------|---|---|---|---|-----|
| ERM | 97.2 ± 0.1 | 66.0 ± 0.6 | 68.9 ± 0.5 | 73.8 ± 0.5 | 76.5 |
| IRM | 97.0 ± 0.0 | 66.3 ± 0.5 | 69.1 ± 0.5 | 74.2 ± 0.2 | 76.7 |
| GroupDRO | 96.9 ± 0.2 | 67.1 ± 0.4 | 69.3 ± 0.2 | 74.6 ± 0.3 | 77.0 |
| Mixup | 97.3 ± 0.1 | 66.7 ± 0.4 | 68.7 ± 0.5 | 73.7 ± 0.1 | 76.6 |
| MLDG | 97.2 ± 0.1 | 63.9 ± 0.6 | 68.6 ± 0.3 | 73.7 ± 0.3 | 75.8 |
| CORAL | 96.9 ± 0.2 | 66.3 ± 0.2 | 68.8 ± 0.5 | 73.9 ± 0.3 | 76.5 |
| VREx | 97.2 ± 0.1 | 66.9 ± 0.0 | 68.4 ± 0.5 | 74.2 ± 0.4 | 76.7 |
| WRI | 97.1 ± 0.2 | 67.0 ± 0.2 | 69.6 ± 0.6 | 74.3 ± 0.2 | 77.0 |

**PACS**

| Algorithm | A | C | P | S | Avg |
|-----------|---|---|---|---|-----|
| ERM | 80.5 ± 0.3 | 81.5 ± 0.2 | 96.6 ± 0.1 | 80.2 ± 0.1 | 84.7 |
| IRM | 80.8 ± 0.9 | 82.0 ± 0.3 | 96.3 ± 0.2 | 79.9 ± 0.3 | 84.7 |
| GroupDRO | 81.4 ± 0.3 | 81.2 ± 0.3 | 96.4 ± 0.2 | 80.2 ± 0.1 | 84.8 |
| Mixup | 80.9 ± 0.3 | 81.7 ± 0.3 | 96.6 ± 0.2 | 80.8 ± 0.2 | 85.0 |
| MLDG | 80.0 ± 0.5 | 81.1 ± 0.4 | 95.7 ± 0.2 | 72.2 ± 0.4 | 82.3 |
| CORAL | 81.4 ± 0.2 | 81.8 ± 0.4 | 96.8 ± 0.1 | 80.5 ± 0.2 | 85.1 |
| VREx | 80.6 ± 0.5 | 81.8 ± 0.7 | 96.8 ± 0.1 | 80.0 ± 0.2 | 84.8 |
| WRI | 81.2 ± 0.3 | 82.3 ± 0.1 | 96.5 ± 0.2 | 80.7 ± 0.1 | 85.2 |

**OfficeHome**

| Algorithm | A | C | P | R | Avg |
|-----------|---|---|---|---|-----|
| ERM | 58.9 ± 0.5 | 50.2 ± 0.3 | 74.4 ± 0.2 | 74.5 ± 0.4 | 64.5 |
| IRM | 57.9 ± 0.6 | 49.8 ± 0.5 | 73.4 ± 0.8 | 74.1 ± 0.8 | 63.8 |
| GroupDRO | 59.7 ± 0.3 | 50.7 ± 0.2 | 74.8 ± 0.1 | 75.4 ± 0.2 | 65.1 |
| Mixup | 61.1 ± 0.2 | 52.2 ± 0.1 | 75.6 ± 0.1 | 75.7 ± 0.1 | 66.1 |
| MLDG | 59.1 ± 0.4 | 50.7 ± 0.2 | 74.8 ± 0.1 | 74.8 ± 0.3 | 64.8 |
| CORAL | 59.3 ± 0.2 | 50.6 ± 0.4 | 74.9 ± 0.2 | 75.2 ± 0.2 | 65.0 |
| VREx | 58.8 ± 0.5 | 50.4 ± 0.3 | 74.4 ± 0.2 | 74.8 ± 0.4 | 64.6 |
| WRI | 58.9 ± 0.6 | 49.8 ± 0.2 | 74.7 ± 0.1 | 74.7 ± 0.3 | 64.5 |

**TerraIncognita**

| Algorithm | L100 | L38 | L43 | L46 | Avg |
|-----------|------|-----|-----|-----|-----|
| ERM | 50.3 ± 0.4 | 50.0 ± 0.6 | 57.8 ± 0.3 | 46.6 ± 0.4 | 51.2 |
| IRM | 53.6 ± 0.7 | 53.2 ± 0.8 | 57.7 ± 0.2 | 46.6 ± 0.8 | 52.8 |
| GroupDRO | 50.4 ± 0.5 | 50.1 ± 0.3 | 57.4 ± 0.3 | 47.3 ± 0.3 | 51.3 |
| Mixup | 53.1 ± 1.8 | 52.4 ± 0.9 | 57.1 ± 0.5 | 46.6 ± 0.3 | 52.3 |
| MLDG | 44.7 ± 0.6 | 49.3 ± 0.3 | 57.2 ± 0.2 | 41.8 ± 0.3 | 48.3 |
| CORAL | 51.0 ± 0.5 | 51.5 ± 1.0 | 57.8 ± 0.2 | 47.0 ± 0.5 | 51.8 |
| VREx | 52.5 ± 1.1 | 51.2 ± 0.6 | 57.9 ± 0.3 | 47.2 ± 0.4 | 52.2 |
| WRI | 51.7 ± 0.5 | 55.0 ± 0.6 | 57.2 ± 0.5 | 47.1 ± 0.3 | 52.7 |

**DomainNet**

| Algorithm | clip | info | paint | quick | real | sketch | Avg |
|---|---|---|---|---|---|---|---|
| ERM | $47.9 \pm 0.2$ | $16.3 \pm 0.1$ | $40.6 \pm 0.2$ | $9.6 \pm 0.2$ | $46.7 \pm 0.2$ | $40.2 \pm 0.1$ | 33.5 |
| IRM | $31.9 \pm 4.4$ | $11.4 \pm 1.3$ | $28.9 \pm 3.0$ | $6.5 \pm 0.7$ | $30.8 \pm 3.9$ | $26.9 \pm 3.7$ | 22.7 |
| GroupDRO | $43.8 \pm 0.3$ | $15.8 \pm 0.2$ | $37.5 \pm 0.3$ | $8.8 \pm 0.1$ | $42.6 \pm 0.2$ | $37.1 \pm 0.3$ | 30.9 |
| Mixup | $47.2 \pm 0.1$ | $16.3 \pm 0.2$ | $40.2 \pm 0.2$ | $9.7 \pm 0.1$ | $45.5 \pm 0.3$ | $39.6 \pm 0.2$ | 33.1 |
| MLDG | $46.9 \pm 0.2$ | $16.2 \pm 0.1$ | $40.0 \pm 0.2$ | $9.2 \pm 0.1$ | $46.2 \pm 0.1$ | $39.4 \pm 0.1$ | 33.0 |
| CORAL | $47.9 \pm 0.3$ | $16.5 \pm 0.1$ | $40.5 \pm 0.2$ | $9.6 \pm 0.1$ | $46.6 \pm 0.2$ | $39.8 \pm 0.2$ | 33.5 |
| VREx | $37.1 \pm 3.2$ | $14.0 \pm 0.9$ | $31.9 \pm 2.8$ | $7.5 \pm 0.7$ | $37.2 \pm 2.5$ | $32.0 \pm 2.5$ | 26.6 |
| WRI | $46.7 \pm 0.2$ | $15.9 \pm 0.1$ | $39.8 \pm 0.1$ | $9.3 \pm 0.1$ | $46.0 \pm 0.2$ | $39.4 \pm 0.1$ | 32.8 |

