# OpenReview forum: "Weighted Risk Invariance: Domain Generalization under Invariant Feature Shift"
_TMLR — Accepted by TMLR_

### Review · Reviewer_svP3 · 2024-04-27

**Summary Of Contributions:**

This paper focuses on domain generalization under the covariate shifts of invariant features. The authors argue that it needs to also consider the density ratios of the underlying invariant features in order to generalize better under the considered distribution shifts. Then, they propose a new framework called weighted risk invariance (WRI), that learns environment weights based on invariant features, and optimizes for invariance under the weighted environment risks. They discuss the difference of WRI with IRM and REx, as well as its implementation. They demonstrate the effectiveness of WRI through experiments on ColoredMNIST and DomainBed.

**Audience:**

Yes

**Claims And Evidence:**

No

**Requested Changes:**

1. Despite a novel problem setting, it's unclear:
- whether the assumptions about the invariant features are realistic. It'd be better to provide some illustrative examples, case studies as well as demonstrations in the benchmarks. For example, do the invariant features really shift in some datasets from DomainBed?
- whether the problem is feasible. It's already been argued that the weights for covariate shifts are not unique. Given that, is there really a solution for the proposed problem setting? Or why can previous approaches for covariate shifts/concept shifts/subpopulation shifts/invariant learning not solve it? Can we solve it without any additional assumptions?

2. There is no guarantee for the solution. Meanwhile, critical details such as how to find the weights (i.e., calculating $R^{e_i,e_j}(f)$) have not been well elaborated in the main text. Since a proper estimation of the weights depends on the identification of the invariant features,
would not the problem be a "the chicken-and-egg conundrum"?

3. Some important related haven't been compared or discussed. For example, [1,2,3,4] also discuss the drawbacks of the invariant learning framework and propose new solutions. [5] generalize the importance weighting to broader distribution shifts.

**References**
[1] Does Invariant Risk Minimization Capture Invariance? AISTATS'21.

[2] Fishr: Invariant Gradient Variances for Out-of-Distribution Generalization, ICML'22.

[3] Pareto Invariant Risk Minimization, ICLR'23.

[4] Understanding and Improving Feature Learning for Out-of-Distribution Generalization, NeurIPS'23.

[5] Generalizing Importance Weighting to A Universal Solver for Distribution Shift Problems, NeurIPS'23.

**Strengths And Weaknesses:**

(+) The problem setting is novel, and is presented clearly.

(+) The discussion and comparison with previous prevalent methods such as REx and IRM are comprehensive.

(+) Empirical studies demonstrate certain advantages of WRI;

(-) Despite a novel problem setting, it's unclear whether it's realistic and feasible;

(-) There is no guarantee for the solution;

(-) Some important related haven't been compared or discussed;

---

> ### Author Response · Authors · 2024-05-11
> **Response to svP3 (1/2)**
>
> Thank you for your review! We appreciate that you consider the problem setting clear and the comparison with baselines comprehensive. Below, we respond to the questions you raised.
>
> **1a. On the invariant covariate shift setting**
>
> We appreciate that you consider our problem setting, as a whole, novel. If we were to break down the setting, we would suggest that it is comprised of both novel and established components. Although the causal structure and the assumption of a constant conditional distribution $p(Y|X_{inv})$ across environments are common in causal invariant learning [6-8], our specific emphasis on invariant covariate shift and our approach to addressing it are, to our knowledge, novel.
>
> > It'd be better to provide some illustrative examples, case studies
>
> As an example of this assumption, we can imagine having data from various hospitals where the demographics change. If we think of smoking as a causal feature for predicting cancer, and the percentage of smokers changes across hospital environments, then this is a case of invariant covariate shift. (Figure 1 shows a case of invariant covariate shift for 2D data. If we match our example to Figure 1, $x_{inv}$ would be some measure of how much someone smokes, and $x_{spu}$ a spurious feature, like text markers or patient positioning in radiographs [9].)
>
> Of course, identifying causal/invariant features is a difficult problem, particularly when we look at high dimensional images. In the canonical example of classifying cows vs camels [10], invariant features could be abstract concepts like the colors of the animals, the existence of horns, etc., and the distribution of these features can shift between domains. For this reason, the invariant features in the real-world image datasets of DomainBed are difficult to isolate for study. We see our real-world DomainBed experiments as evidence that our method is competitive on high dimensional, real-world images, even though it is hard to verify whether this assumption holds in high dimensional data.
>
> > It’d be better to provide some…demonstrations in the benchmarks
>
> CMNIST is a DomainBed dataset with a clear spurious feature (color) and a clear invariant feature (digit) [7]. We created invariant covariate shift in CMNIST by sampling the digits of each environment from different distributions. We use our performance on this benchmark to demonstrate that our method is robust to shift.
>
> **1b. On the feasibility of invariant feature weighting**
>
> As you point out, our weighting functions are not unique, they only have to fit a specific constraint. This means that there are multiple reweightings that align the invariant features, including density weighting and density ratio weighting (Definition 2). These reweightings all allow us to solve for an invariant predictor, and in that sense, can all be considered “solutions” to the invariant feature weighting problem (although some choices may have benefits over others, see Appendix B).
>
> Previous approaches for covariate shift/concept shift/subpopulation shift study shifts in the input data or joint features. However, they do not distinguish the invariant and spurious features, and also do not aim to correct for shifts in the invariant features specifically.
>
> Previous approaches for invariant learning do distinguish between the invariant and spurious features, and we specifically compare to other invariant learning works, including IRM and VREx, that study the setting where there are invariant features $X_{inv}$ such that $p(Y|X_{inv})$ is constant across environments. As we explore more in the paper, we see that these methods also have drawbacks when it comes to finding an invariant predictor.
>
> **References ([1-5] follow references from review)**
>
> [6] Scholkopf et al, On Causal and Anticausal Learning (2012)
>
> [7] Arjovsky et al, Invariant Risk Minimization (2019)
>
> [8] Krueger et al, Out-of-Distribution Generalization via Risk Extrapolation (2021)
>
> [9] DeGrave et al, AI for radiographic COVID-19 detection selects shortcuts over signal (2021)
>
> [10] Beery et al, Recognition in terra incognita (2018)

---

> ### Author Response · Authors · 2024-05-11
> **Response to svP3 (2/2)**
>
> **2. On estimating the weights**
>
> There are two types of guarantees that could be expected: a statistical or an optimization guarantee. We prove a statistical guarantee in Section 3. However, an optimization guarantee is more difficult, as weighting functions that align the invariant features do depend on the invariant features.
>
> Nevertheless, we find that it is possible to estimate these weighting functions without explicit identification of the invariant features. In our experiment in Appendix C, we show that, even when we weight with the density of the joint features (both invariant and spurious—we have access to these features), our WRI objective tends to discard the spurious features, so that we end up with weighting functions that only depend on the invariant features. Then, as we proved in the paper, weighting functions that only depend on the invariant features allow us to learn an invariant predictor. For high-dimensional data, where feature density is notoriously difficult to estimate, we use alternating minimization to train a prediction model and density estimators. Our expectation is the same: that even though we start with density estimation on the joint features, the WRI objective leads us to eventually discard the spurious features. We corroborated this on higher dimensional data and nonlinear models (from the synthetic experiment in Appendix E.2, to CMNIST, to DomainBed).
>
> **3. On related work**
>
> Thank you for pointing out related works [1-5]. Below, we discuss the problems studied in these works, and describe how they relate to our paper.
>
> - We reference [1] at the start of “Comparison to IRM” in Section 3.3.
> - Fishr [2] studies a different line of invariant learning than what we study in our paper. In Section 2 of [2], the authors discuss a branch of work that is “motivated by arguments from causality”, including IRM and VREx, and they distinguish Fishr as following another line of work on promoting agreements between gradients w.r.t. network weights. Our work is a form of causally-motivated invariant learning, so we compare to IRM and VREx, and we do not directly compare to [2].
> - [3] points out the potential conflict between ERM and OOD objectives, and proposes a new optimization method that allows for cooperation between multiple OOD objectives. Our work also uses the combination of an ERM and OOD objective, and studies the same OOD objectives as the reference (VREx and IRMv1), so this work is very relevant to the optimization of our proposed objective. We thank the reviewer for bringing this work to our attention, and we have added it our Discussion section.
> - Similarly, [4] studies the interplay between ERM and OOD objectives, and proposes a training method that improves the OOD performance of various OOD objectives. Like [3], their results pair very well with our work and can be used to improve training. We have also added this work to our Discussion section.
> - As you describe, [5] characterizes when importance weighting (IW) succeeds or fails, and proposes a generalized form of IW (GIW) that works for a wide array of shifts. Our work applies IW to the invariant learning problem, and may therefore share the limitations of IW for invariant feature shifts. We believe GIW would pair well with our work, so we have added it to our Related Works (“Importance weighting methods”) and Discussion.
>
> Following TMLR guidelines, we will upload an updated manuscript after all 3 reviews have been posted and addressed.

---

> > ### Comment · Reviewer_svP3 · 2024-06-02
> >
> > Thank you for the detailed rebuttal, which solves most of my concerns. Nevertheless, I am still a bit confused by the feasibility.
> >
> > The theory in the paper shows that the weighting functions need to depend only on the invariant features in order to learn an invariant predictor. However, one could not know what the invariant features are before obtaining the invariant feature. How does the proposed solution specifically mitigate the issue?

---

> > > ### Author Response · Authors · 2024-06-04
> > >
> > > To further extend your statement, our theory says that weighting functions $\alpha$ and $\beta$ that obey $\alpha(x_{inv}) p_{e1}(x_{inv}) = \beta(x_{inv}) p_{e2}(x_{inv})$ allow us to optimize an objective that leads to an invariant predictor. In particular, we discuss and show additional results for invariant feature density weighting functions and for invariant feature density ratio weighting functions.
> > >
> > > Naturally, there is the question of how we use invariant feature density ratio weighting functions (for example), before we have access to the invariant features. Motivated by our theory, we propose an alternating minimization method that weights by the *observed, joint* features at each step. We do not provide any guarantees for this method, but Appendices C and D show this claim empirically for the 2D Gaussian case. In Figure C.1, we show that the weight placed on the spurious feature goes to zero while learning with WRI. The plots in Figure D.3 and Figure D.4 show that we learn the invariant feature densities, even though we do not know them before training and learning is performed on the observed features. Further empirical results in controlled settings like CMNIST give additional evidence for this claim for higher dimensions.
> > >
> > > For additional intuition on why this happens, consider trying to reweigh a representation $\phi(X)$ that is *not* invariant (i.e for 2 environments $p_1(Y \vert \phi(X)) \neq p_2(Y \vert \phi(X))$), and let us explain why WRI will reject it. For further simplicity let us focus on density ratio reweighting, and consider the reweighted distribution $\tilde{p} = p_1(\phi(X)) \cdot p_2(Y \vert \phi(X)))$. To satisfy invariance under this reweighting, we must have $\tilde{p}(Y \mid \phi(X) ) = p_1(Y \mid \phi(X))$. Clearly this does not hold, since $\tilde{p}(Y \vert \phi(X)) = p_2(Y \vert \phi(X))$, and this lack of invariance can be identified from data under mild assumptions on overlap. Therefore, enforcing this type of *weighted invariance* constraint (i.e. matching of the conditionals $p_1(Y \vert \phi(X))$ and the density ratio weighted distribution $\tilde{p}(Y \vert \phi(X))$) would rule out a non-invariant representation $\phi(X)$, while it is easy to see that this constraint is satisfied by an invariant representation.
> > >
> > > Now, the gap between this constraint and weighted *risk* invariance is the fact that we only enforce invariance of the risk and not the full conditionals (we do this because imposing risk invariance is more tractable in practice). The two are not equivalent, but there are cases where the notions coincide, e.g. see Theorems 1 and 2 about linear models in the REx paper [1], and their experiments that support the claim that risk invariance learns an invariant predictor.
> > >
> > > A subtle but important point to notice is that these theorems presuppose the existence of a risk invariant predictor. Under the invariant-covariate shift we consider, some of the invariant features have to be discarded in order to obtain a risk-invariant predictor (e.g. in the 2D Gaussian case, a risk invariant predictor would have to discard the causal feature completely). With *weighted* risk invariance this is alleviated, and using e.g. density ratio reweighting, the optimal invariant predictor becomes risk invariant again.
> > >
> > > We did not include this explanation in the paper as it is a bit involved, and we thought the empirical demonstration is a bit more intuitive. If the reviewer finds the above explanation helpful, then we are happy to include it in the appendix as well.
> > >
> > > [1] Krueger et al, OOD generalization via risk extrapolation. 2020

---

> > > > ### Comment · Reviewer_svP3 · 2024-06-12
> > > >
> > > > Thank you for the explanation. I feel it necessary as one of the major contributions in this work is to propose a new problem setting, so the feasibility of the proposed problem is worth a in-depth discussion.

---

> > > > > ### Author Response · Authors · 2024-06-13
> > > > >
> > > > > We appreciate your feedback. We have now included this discussion in Appendix D.1 of our latest revision. Thank you for your suggestion!

---

### Review · Reviewer_d8hB · 2024-04-28

**Summary Of Contributions:**

This paper proposes a new algorithm for domain generalization, called weighted risk invariance (WRI). The objective is to minimize some type of variance after reweighting with the environment distributions.

**Audience:**

Yes

**Broader Impact Concerns:**

The broader impact is properly discussed in Sec 6.

**Claims And Evidence:**

No

**Requested Changes:**

See above.

**Strengths And Weaknesses:**

Strengths:
1. This paper is well written with clear structures. On the theoretical side, it is explicit about the assumptions, such as causal prediction coupling and spurious-free predictor.
2. It compares with VREx and IRM and discuss the similarity and difference.
3. Based on weighted invariance this paper proposes a new loss function and compares the algorithm on variant benchmarks such as Colored MNIST and DomainBed.

Weaknesses:
1. I still have some questions regarding the assumptions. As mentioned in the paper, most common assumptions in DG are covariant shifts, label shifts, and invariant predictors. However, this paper is based on a new assumption called causal prediction coupling. Why is this assumption valid and practical? Or could this be the reason why it doesn't perform significantly better in practice? Also, the linear causal setting seems limited as well. How does WRI perform in non-linear settings?
2. Figure 1 seems confusing. Is this figure to demonstrate that WRI is better than other algorithms, and in which sense?
3. Although Prop. 1 seems intriguing for a new principle of domain invariance, it might be a weak condition. As the authors mentioned, a trivial solution can be $\alpha = \beta = 0$. Is there a way to say that the WRI condition is tight or sufficient as well (in general)? Or could the authors share some intuition?
4. In order to enforce WRI, in eq. (6) the authors need to minimize all pairs of environments. This quadratic scaling makes it infeasible for a large number of envs. Could the authors show some comparison of the computation time with a large number of environments?
5. Sec 3.3 only compares with REx and IRM in some special examples, but fails to prove the advantage in general.
6. In Sec 3.4, a log density loss was added ($\log d^e$). Could the authors explain why it is necessary or theoretically justify it? Some convergence proof might be helpful.
7. The main experiments only compare with IRM, ERM and VREx. If we look at the appendix, WRI is in fact not better than simpler algorithms like CORAL. This minor improvement might show that the underlying assumption (causal prediction coupling) does not hold in general.

---

> ### Author Response · Authors · 2024-05-11
> **Response to d8hB (1/2)**
>
> Thank you for your review! We appreciate that you consider the paper well written and well structured, with explicit theoretical contributions. We respond to your questions below.
>
> **1. On the causal prediction coupling assumption**
>
> To clarify, the causal structure we study, and the accompanying assumption (Assumption 1), appear often in works on causally-motivated invariant learning [1-3]. As an example of this assumption, we can imagine having data from various hospitals where the demographics change. If we think of smoking as a causal feature for predicting cancer, and the percentage of smokers changes across hospital environments, then this is a case of invariant covariate shift.
>
> The linear causal setting is amenable to analysis; for this reason, existing literature in this area provides theoretical justification for either the linear case [2-4] or for highly specific nonlinear distributions under additional assumptions [5]. Thus, we ground our theory in the linear causal setting, but we extend our experiments to nonlinear settings as well. We first provide theoretical and empirical evidence for the linear model, then empirical evidence for the nonlinear model where some version of our causal structure holds (CMNIST), and finally empirical evidence on DomainBed, a benchmark of unstructured datasets with unknown shifts. In this way, we hope to form a coherent narrative from our theory (based on the linear causal setting) to experiments beyond that.
>
> **2. On Figure 1**
>
> The top row of Figure 1 shows a Gaussian setup where there exist invariant features $x_{inv}$ such that $p(y|x_{inv})$ is the same across environments, and spurious features $x_{spu}$ such that $p(y|x_{spu})$ changes across environments (following our causal model). The environments exhibit invariant covariate shift, as the distribution of invariant features $x_{inv}$ changes across environments. Under this setting, we demonstrate that the WRI objective recovers a more “invariant” predictor than the other algorithms (in the bottom row of the figure). That is, the learned predictor for WRI bases its predictions more on the invariant features $x_{inv}$ and less on the spurious features $x_{spu}$, so it learns a decision boundary that is vertical in this case—it only changes with $x_{inv}$, not $x_{spu}$.
>
> **3. On Proposition 1**
>
> Proposition 1 establishes that a predictor being spurious-free is a sufficient condition for WRI to hold. However, it also admits trivial solutions that allow predictors to satisfy WRI without being spurious-free. To make our theory more meaningful, we also prove a result in the opposite direction: that under general position, for the case of linear regression, a predictor that satisfies WRI is spurious-free (Theorem 2). The general position assumption does not allow for solutions like $\alpha=\beta=0$; thus, Proposition 1 and Theorem 2, with the additional assumptions mentioned, establish that WRI is necessary and sufficient for spurious-free prediction.
>
> **4. On quadratic scaling**
>
> While we minimize over all pairs of environments in the general case, for a general choice of weighting functions, we note that the number of constraints can be reduced depending on the choice of weighting function. For example, if we use density ratio weighting, then we can set a reference environment and weight all the other environments according to the weight on the reference environment (this is the intuition behind equation A.9). That means for density ratio weighting, if we have $k$ environments, then we only need to minimize over $k$ constraints. As a comparison, this is the same number of constraints as is needed for IRM.
>
> We thank the reviewer for pointing this out and we added details to the revision that hopefully make this more clear.
>
> **References**
>
> [1] Scholkopf et al, On Causal and Anticausal Learning (2012)
>
> [2] Arjovsky et al, Invariant Risk Minimization (2019)
>
> [3] Krueger et al, Out-of-Distribution Generalization via Risk Extrapolation (2021)
>
> [4] Peters et al (2015). Causal inference using invariant prediction
>
> [5] Dong and Ma (2023). First Steps Toward Understanding the Extrapolation of Nonlinear Models to Unseen Domains

---

> ### Author Response · Authors · 2024-05-11
> **Response to d8hB (2/2)**
>
> **5. On the benefits of WRI over REx/IRM**
>
> We want to clarify that our comparisons to REx and IRM are not intended to be special examples, in the sense that they are rare cases, but should illustrate problems that do arise in realistic settings. For example, REx fails under heteroskedasticity and invariant covariate shift—a common case in heterogeneous data—and WRI is robust to this case. In Section 3.3., we break down these cases to provide more explanation/evidence of why these failure modes occur. Afterwards, we show improvement in more general settings, working our way up to high-dimensional examples. First, we show that WRI performs well on Gaussian data, ranging from 2D (Figure 1) to 20-dimensional (Appendix E.2). Then, we show improvement in CMNIST, a popular benchmark with a nonlinear structure. Finally, we show competitive performance on DomainBed, a benchmark with unknown shifts.
>
> We hope that this answers the reviewer’s concerns. If there are any remaining questions, we would be happy to discuss what type of evidence for general advantage that the reviewer is looking for.
>
> **6. On the log density term**
>
> In our approach, we perform density weighting on a learned representation. A possible failure mode would be a representation where there is no overlap between environments, so that $\alpha=\beta=0$, as you mention above. There are different ways to prevent this from happening, but one way that we find effective is to use a log density term. The log density term encourages high entropy and a large support for each environment, and we observe empirically that it encourages overlap between environments. We thank the reviewer for raising this point, and we have added this explanation to our revision.
>
> We recognize the utility of a convergence proof here, but we also find this problem to be very complex, especially for a nonconvex setting like ours. We will highlight this as an area for future research in our revision.
>
> **7. On main experiments**
>
> WRI is a form of causally-motivated invariant learning, so we prioritize comparisons with causally motivated methods like IRM and VREx. Causally-motivated methods have the benefit of interpretability and sound theoretical grounding. As a trade off, they can be difficult to scale to complex, high dimensional data. Overall, we believe that pushing forward the state of the art in causally-motivated methods is, in itself, a significant contribution.
>
> As you note, WRI has the same accuracy as CORAL on DomainBed. However, WRI also offers distinct advantages: our objective not only provably discards spurious features, but we also learn a density estimate of the invariant features.
>
> Following TMLR guidelines, we will upload an updated manuscript after all 3 reviews have been posted and addressed.

---

### Review · Reviewer_RbPh · 2024-05-13

**Summary Of Contributions:**

This submission proposes a method for learning invariant representations in the presence of covariate shift. A causal data generation process is proposed and an idealised objective is derived from this. A heuristic agorithm for minimising this objective given only a finite sample is prposed, and experimentally validated on some domain generalisation problems. There are also some additional experiments that provide some corroboration to other claims made along the way.

**Audience:**

Yes

**Broader Impact Concerns:**

The work is mainly theoretical in general, but the topic (robust generalisation) is more likely to lead to a positive broader impact than negative, in my view. The authors include some discussion on broader impact concerns in their paper.

**Claims And Evidence:**

No

**Requested Changes:**

* Provide an example problem where the causal structure assumed in this work is present.
* Provide an intuitive example to compare and contrast the relationship of invariant covariate shift with regular covariate shift.
* Explain why we should expect optimising the proposed objectives in either of Equation 6 or Equation 9 to result in accurate feature densities. Alternatively, be upfront that there are no guarantees and quantify experimentally the extent to which alternating optimisation on the heuristic density estimation objective does find good approximations of the (un-normalised) densities.
* Fix the sample complexity results in Appendix B, or remove them.
* State more clearly some of the assumptions about how different environments must be related in order for this method to work. E.g., should all environments have the same support, as is typically required for density ratio methods?

**Strengths And Weaknesses:**

## Strengths:
* The writing is generally quite good. The paper provides a lot of extra information in the appendix, and a good amount of effort has gone into relating the what has been proposed in this paper with previous work.
* The experiments are focused on understanding rather than trying to beat benchmarks, and as a result the paper is quite interesting to read.

## Weaknesses:
* It's difficult to see how the specific causal structure assumed in this work maps onto real-world situations. The motivation and scope of the paper would be easier to understand if the reader was given an example problem where this structure is present. I would also be interested in an intuitive example to compare and contrast the relationship of invariant covariate shift with regular covariate shift.
* It's not clear why we should expect optimising the proposed objectives in either of Equation 6 or Equation 9 to result in accurate feature densities. The derivation only explains how one can obtain robust models if one already has accurate densities, not the other way around. The experimental results (Table 3) indicate that the learned densities, while more useful that other methods, are still of low quality.
* Appendix B does not provide sample complexity results; in order to apply Bernstein's inequality in this context one must assume that the model is independent of the training data, which is not true when the data is used to train the model. Frameworks based on, e.g., covering numbers have been developed to get around this. The union bound should also be used to combine the two applications of Bernstein inequality used to obtain B.14, since these two separate bounds only hold with high probability. Assuming this is fixed, it should also be made explicit that this convergence rate does not necessarily apply to the proposed algorithm, since this bound assumes optimal weighting functions are known. Moreover, the conclusion that one should select densities with lower maximum values and lower variances does not make sense; surely the ideal densities are the true densities of the underlying data distributions? Also, if the densities are probability densities, reducing the maximum would require increasing the variance in order for the density to remain normalised.

---

> ### Author Response · Authors · 2024-05-23
>
> Thank you for your review! We did try to focus on exploring the problem setting over performance on benchmarks, and we are grateful that this intention was evident in your reading of the paper. We also appreciate your clear translation of concerns to requested changes. Below, we discuss how we incorporated these changes in our revision.
>
> **1. Example Problem with Causal Structure**
>
> We added a real-world example of our causal structure to Section 2.3. In our example, smoking is a causal/invariant feature $X_{inv}$ for predicting the presence of cancer $Y$. Suppose we have data from various hospitals, and the percentage of smokers varies across hospitals. This setting is a case of invariant covariate shift because while $p(X_{inv})$ changes across hospitals, we assume that the conditional distribution $p(Y|X_{inv})$ remains the same.
>
> **2. Intuitive Example Comparing Invariant Covariate Shift with Regular Covariate Shift**
>
> We call this “invariant covariate shift” because we want to emphasize that this refers to covariate shift in the invariant features specifically. In contrast, regular covariate shift applies to the entire input, not just specific features. For example, if the entire input is all patient demographics $X$, then regular covariate shift assumes that the conditional distribution of cancer given all patient demographics $p(Y|X)$ is the same across hospitals. We consider our assumption that $p(Y|X_{inv})$ is the same across hospitals to be more targeted and realistic.
>
> **3. Experimental results on recovering feature densities**
>
> Given that our practical alternating optimization objective is a hard non-convex optimization problem, it is hard to provide any optimization guarantees. Following the reviewer's guidance, we make this explicit in Section 3.4 of the manuscript.
>
> However, we have conducted the recommended experiments to show that alternating optimization does approximate the feature densities in a 2D synthetic case. The results of these experiments have been included in Appendix D.3. While we do not provide formal guarantees, we hope that these results offer some evidence to support the effectiveness of our approach. We thank the reviewer for their suggestion, which helped us improve the clarity and robustness of our paper.
>
> **4. Fixing the sample complexity results in Appendix B**
>
> We have made several updates to Appendix B following your guidance. Instead of a trained model $f$, we now fix $f$ as a hypothesis in some hypothesis set $H$. This is the same assumption that is made by Cortes [1], whose results we are extending; as with Cortes, we hope the results in Appendix B can still act as a motivating exploration on the sample complexity of weighted risk under specific assumptions. However, since these assumptions differ from our setting (where we use a trained model $f$), we make this difference explicit in the Appendix. We also make explicit that our bound assumes the optimal weighting functions are known, and that it is not the case for our proposed algorithm. Because of these assumptions, we only put this motivating analysis in a footnote.
>
> We corrected our work to use the union bound to combine the two applications of Bernstein’s inequality.
>
> Finally, we want to clarify that we are not suggesting that we should select densities with lower maximum values and variances, but *weighting functions* with lower maximum values and variances. These bounds may suggest that density weighting functions ($\alpha = p_i(x_{inv})$, $\beta = p_j(x_{inv})$) are preferable to density ratio weighting functions ($\alpha = 1$, $\beta = p_j(x_{inv})/p_i(x_{inv})$), which often have larger maximum values/variances.
>
> **5. Relation of different environments**
>
> There are multiple choices of weighting function that allow for weighted risk invariance, and assumptions about how the different environments must be related depend on the choice of weighting function. Density ratio weighting, where we weight all environments to match a selected reference environment, would require that the support of that reference environment be contained within the support of each of the other environments—otherwise, the weighted risk objective would be misaligned. In contrast, density weighting does not impose this requirement on the support of the environments, and allows for invariant learning over all regions of invariant feature support overlap. Note that some support overlap is necessary for invariant learning to meaningfully occur; this is a fundamental limitation of invariant learning in general [2], and violating this does not misalign the weighted risk objective. In non-overlapping regions, the weighted risk would simply be zero, causing the ERM objective to take over.
>
> We thank the reviewer for bringing up this important point, and we have added this description to Section 3.2.
>
> **References**
>
> [1]Cortes, Learning Bounds for Importance Weighting (2010)
>
> [2]Ahuja, Invariance Principle Meets Information Bottleneck (2021)

---

### Author Response · Authors · 2024-05-23
**Global response**

We want to sincerely thank all reviewers for their time, engagement, and thoughtful reviews. In this global response, we want to summarize the changes we made in our revision following reviewer feedback.

- We added an illustrative example of the problem setting in real world situations to Section 2.3 (svP3, d8hB, RbPh). We also clarify the difference between invariant covariate shift and regular covariate shift  in the context of this example (RbPh).
- We emphasize that our practical objective (eq. 9) is a heuristic without convergence guarantees (RbPh), and we add experiments to help readers understand how effective alternating optimization is at approximating the invariant feature densities to Appendix D.3 (RbPh).
- We highlight the optimization process as an area for future research in the Discussion (d8hB).
- We made several fixes to the sample complexity results in Appendix B, according to reviewer suggestions (RbPh).
- We added clarifications on the number of constraints needed for WRI in Section 3.2 (d8hB).
- We provided some intuition for the log density term in eq. 9 in Section 3.4 (d8hB).
- We added a discussion on how different choices of weighting function lead to different requirements on the support of the environments in Section 3.2 (RbPh).
- We included additional related work in the Related Work and Discussion sections (svP3).
- We updated the caption for Figure 1 to better explain the setting and result (d8hB).

Changes between the original submission and revision are written in red for visibility.

---

### Decision · Action_Editor_nQ4s · 2024-06-28

**Recommendation:** Accept with minor revision

**Comment:**

This work combined two directions in domain generalization: covariate shift and invariant risk minimization, by allowing the invariant features to go through a (marginal) distribution shift among different domains (environments). This new problem setting, albeit a bit straightforward, could be interesting in some applications. The proposed algorithm is also a combination of existing ideas in covariate shift (density weighting) and invariant risk minimization (variance regularization). The chicken-egg problem raised by reviewer svP3 was addressed by alternating minimization, which, albeit lack of formal guarantee, seems to be a reasonable workaround and shows some promise in the authors' experiments. Overall the reviewers agreed that the problem setting is interesting. There are some doubts on the comparison against stronger baselines such as CORAL and GroupDRO, but I think the authors have done reasonable justification of their methodology through experiments on both synthetic examples and real datasets.

Some minor suggestions:

(a). For Eq (9), instead of saying the last log term encourages high entropy (why is this a good thing?), why not just point out we are performing maximum likelihood density estimation for each environment (in the feature space)? It would be nice to comment on the role of each term in Eq (9). For instance, without the first ERM term, the result could be trivial?

(b). In comparison to VREx (7), it would be nice to point out the proposed method in Eq (6) is exactly Var(R^{e_i}(f)), so the main difference simply lies in the weighting in the risk R^{e_i}(f): In particular, if one uses uniform weighting we recover VREx. The authors seem to know this: the reply to Reviewer d8hB (On quadratic scaling) perhaps is also worth incorporating into the final version.

(c). "In Appendix C, we show that ... WRI on the joint features (rather than the invariant features) still leads to learning invariant feature weighting functions." Appendix C is an experiment on a toy dataset, so I would suggest changing "show" to "empirically show" in this claim.

**Audience:**

Yes.

**Claims And Evidence:**

Yes, the claims are supported by experiments and some proofs in the appendix.

---

> ### Author Response · Authors · 2024-07-25
>
> We thank the editor for their valuable feedback!
>
> We have uploaded a revision that incorporates suggestions (b) and (c) directly. Following suggestion (a), we have updated our description of the WRI objective following eq. 6, and refer to this objective after our practical objective in eq. 9 to emphasize the connection between the two. We also describe the log term as encouraging maximum likelihood estimation of the density parameters when the feature density family is known. However, if we are dealing with settings where the feature density family is unknown, the behavior of the log term is harder to characterize. We hope the high entropy explanation accommodates this general setting.